# Bayesian Inference for Sequence Mixture Density Networks using Bézier Curve Gaussian Processes

## Abstract

Probabilistic models for sequential data are the basis for a variety of applications concerned with processing timely ordered information. The predominant approach in this domain is given by recurrent neural networks, implementing either a transformative approach (e.g. Variational Autoencoders or Generative Adversarial Networks) or a regression-based approach, i.e. variations of Mixture Density networks (MDN). In this paper, we focus on the $\mathcal{N}$-*MDN* variant, which parameterizes (mixtures of) probabilistic Bézier curves ($\mathcal{N}$-*Curves*) for modeling stochastic processes. While MDNs are favorable in terms of computational cost and stability, they generally fall behind contending approaches in terms of expressiveness and flexibility. Towards this end, we present an approach for improving MDNs in this regard by enabling full Bayesian inference on top of $\mathcal{N}$-MDNs. For this, we show that $\mathcal{N}$-Curves are a special case of non-stationary Gaussian processes (denoted as $\mathcal{N}$-GP) and then derive corresponding mean and kernel functions for different modalities. Following this, we propose the use of the $\mathcal{N}$-MDN as a data-dependent generator for $\mathcal{N}$-GP prior distributions. We show the advantages granted by this combined model in an application context, using human trajectory prediction as an example.

## 1 Introduction

Models of sequential data play an integral role in a range of applications related to representation learning, sequence synthesis and prediction. Thereby, with real-world data often being subject to noise and detection or annotation errors, probabilistic models are favorable. These take uncertainty in the data into account and provide an implicit or explicit representation of the underlying probability distribution.

The determination of such a probabilistic sequence model is commonly layed out as a learning problem, learning a model of an unknown underlying stochastic process from given sample sequences, which are assumed to be realizations of this process. Extending on this by focusing on (multi-step) sequence prediction as an exemplary task, a machine learning model is tasked to predict a distribution over a given number of subsequent sequence elements, given some input sequence. Common approaches are based on either Gaussian Processes (Rasmussen & Williams, 2006) (e.g. Damianou & Lawrence (2013); Mattos et al. (2016)) or more prevalently on neural networks either using stochastic weights (Bayesian Neural Networks (*BNN*) (Bishop, 1995; Blundell et al., 2015; Gal & Ghahramani, 2016)), stochastic input (transformative[1] neural models, e.g. (conditional) Variational Autoencoders (*cVAE*, Kingma & Welling (2014); Sohn et al. (2015); Bowman et al. (2016)) and (conditional) Generative Adversarial Networks (*cGAN*, Goodfellow et al. (2014); Mirza & Osindero (2014); Yu et al. (2017))) or parameterizing mixture distributions (Mixture Density Networks (*MDN*) (Bishop, 1994; Graves, 2013)), combined with recurrent neural networks for handling sequential data in an autoregressive manner. In general, the neural models containing stochastic components allow to directly sample from the modeled stochastic process. For this, these models typically require computationally expensive Monte Carlo methods during training and inference. In contrast, MDN-based models are deterministic and map a given input onto the parameters of a mixture distribution. While these

---

[1]Transformative approaches take samples of a simple probability distribution (e.g. unit Gaussian) and transform them into a sample-based representation of a more complex probability distribution.

models are generally more stable and less computationally expensive during training, Monte Carlo methods are still required for multi-modal inference. Further, due to MDN-based models being simpler models, they are potentially less expressive and flexible.

In order to tackle difficulties with multi-modal inference in MDN-based probabilistic sequence models, Hug et al. (2020) proposed a variation of MDNs, which operate in the domain of parametric curves instead of the data domain, allowing to infer multiple time steps in a single inference step. The model is built on a probabilistic extension of Bézier curves ($\mathcal{N}$-*Curves*), which assume the control points to follow independent Gaussian distributions, thus passing stochasticity to the curve points. Following this, the $\mathcal{N}$-*MDN* generates a sequence of Gaussian mixture probability distributions in terms of a mixture of $\mathcal{N}$-Curves.

Extending on this approach, in this paper we enhance MDNs for probabilistic sequence modeling in terms of flexibility and expressiveness by establishing a connection between $\mathcal{N}$-MDNs and the Gaussian process (GP) framework. Our basic idea revolves around employing the $\mathcal{N}$-MDN for determining a data-dependent GP prior based on $\mathcal{N}$-Curves, which enables the manipulation of predictions generated by an $\mathcal{N}$-MDN in a post-hoc manner. To achieve this, we first show that the underlying $\mathcal{N}$-Curves are a special case of Gaussian processes. We denote this $\mathcal{N}$-Curve – induced GP as $\mathcal{N}$-GP in order to reduce ambiguity[2]. Following this, we derive mean and kernel functions for the $\mathcal{N}$-GP considering different cases, i.e. univariate, multivariate and multi-modal. Ultimately, this allows for a more expressive and flexible probabilistic sequence model by employing the GP framework for enabling Bayesian posterior inference, with the benefits of a regression-based neural network model for parameter estimation.

In our evaluation, we explore the advantages granted by our combined model from a practical perspective. Following this, using human trajectory prediction as an exemplary sequence predicton task, we use our model for manipulating predictions generated by the underlying $\mathcal{N}$-MDN by calculating different posterior distributions according to the induced $\mathcal{N}$-GP. For the posterior distributions, we consider two use cases. First, improving the overall prediction performance by calculating the posterior predictive distribution given one or more observed trajectory points. Second, we explore the possibilities of updating the predictions under the presence of new measurements within the predictive time horizon. In our approach, this does not require any additional passes through the $\mathcal{N}$-MDN.

To summarize, our main contributions are given by:

1. A proof for probabilistic Bézier curves ($\mathcal{N}$-Curves) being Gaussian processes.

2. The derivation of GP mean and covariance functions induced by $\mathcal{N}$-Curves, covering the univariate, multivariate and multi-modal cases.

3. A probabilistic sequence model, which combines the stability and low computational complexity of $\mathcal{N}$-Curve – based MDNs with the expressiveness and flexibility of Gaussian processes.

The paper is structured as follows. Section 2 gives a brief introduction to Gaussian processes, probabilistic Bézier curves and Mixture Density Networks, providing the basis for this paper. In section 3, we provide an equivalence proof for $\mathcal{N}$-Curves and Gaussian processes and derive $\mathcal{N}$-GP mean and covariance functions for the univariate, multivariate and multi-modal case. In order to make the paper more self-contained and easy to follow, the corresponding subsections for each case also include a problem description and a brief overview of prior work relevant for the derivation of the respective $\mathcal{N}$-GP mean and kernel functions in addition to our $\mathcal{N}$-GP derivation itself. In section 4 we then establish a link between $\mathcal{N}$-MDNs and $\mathcal{N}$-GPs and present use cases for the combined model we consider relevant in practice. Next, section 5 provides a brief overview of other related areas of research, complementing the references given in section 3. Finally section 6 investigates on the viability of our combined model and section 7 concludes the paper.

---

[2]We will use the term $\mathcal{N}$-Curve exclusively to refer to the polynomial curve with variance in its control and curve points and $\mathcal{N}$-GP for referring to the actual Gaussian process, which can be calculated from an $\mathcal{N}$-Curve.

## 2 Preliminaries

### 2.1 Gaussian Processes

A Gaussian process (GP, Rasmussen & Williams (2006)) is a stochastic process $\{X_t\}_{t \in T}$ with index set $T$, where the joint distribution of stochastic variables $X_{t_i}$ for an arbitrary, finite subset $\{t_1, ..., t_N\}$ of $T$ is a multivariate Gaussian distribution. Throughout this paper, the index will be interpreted as *time*. The joint distribution is obtained using an explicit mean function $m(t)$ and positive definite covariance function $k(t_i, t_j) = \text{cov}(f(t_i), f(t_j))$, commonly referred to as the kernel of the Gaussian process, and yields a multivariate Gaussian prior probability distribution over function space. Commonly, $m(t) = 0$ is assumed. Given a collection of sample training points $X_*$ of a function $f(t)$, the posterior (predictive) distribution $p(X|X_*)$ over non-observed function values $X$ can be obtained. As such, Gaussian processes provide a well-established model for probabilistic sequence modeling.

### 2.2 Probabilistic Bézier Curves

Probabilistic Bézier Curves ($\mathcal{N}$-Curves, Hug et al. (2020)) are Bézier curves (Prautzsch et al., 2002) defined by $(L+1)$ independent $d$-dimensional Gaussian control points $\mathcal{P} = \{P_0, ..., P_L\}$ with $P_l \sim \mathcal{N}(\boldsymbol{\mu}_l, \boldsymbol{\Sigma}_l)$. Through the curve construction function

$$X_t = B_{\mathcal{N}}(t, \mathcal{P}) = \mathcal{N}(\mu_{\mathcal{P}}(t), \Sigma_{\mathcal{P}}(t)) \tag{1}$$

with

$$\mu_{\mathcal{P}}(t) = \sum_{l=0}^{L} b_{l,L}(t) \boldsymbol{\mu}_l \tag{2}$$

$$\Sigma_{\mathcal{P}}(t) = \sum_{l=0}^{L} (b_{l,L}(t))^2 \boldsymbol{\Sigma}_l, \tag{3}$$

where

$$b_{l,L}(t) = \binom{L}{l}(1-t)^{L-l}t^l \tag{4}$$

are the Bernstein polynomials (Lorentz, 2013), the stochasticity is passed from the control points to the curve points $X_t \sim \mathcal{N}(\mu_{\mathcal{P}}(t), \Sigma_{\mathcal{P}}(t))$, yielding a sequence of Gaussian distributions $\{X_t\}_{t \in [0,1]}$ along the underlying Bézier curve. Thus, a stochastic process with index set $T = [0, 1]$ can be defined. For representing discrete data, i.e. sequences of length $N$, a discrete subset of $T$ can be employed for assigning sequence indices to evenly distributed values in $[0, 1]$, yielding

$$T_N = \left\{ \frac{v}{N-1} | v \in \{0, ..., N-1\} \right\} = \{t_1, ..., t_N\}. \tag{5}$$

### 2.3 Mixture Density Networks

A Mixture Density Network (MDN, Bishop (1994)) is a feed-forward neural network

$$\Phi(\mathbf{v}) = (\{\pi_k, \boldsymbol{\mu}_k, \boldsymbol{\Sigma}_k\}_{k \in \{1, ..., K\}} | \mathbf{v}), \tag{6}$$

that takes an input vector $\mathbf{v}$ and maps it onto the parameters of a $d$-dimensional, $K$-component Gaussian mixture distribution. In order to ensure that the MDN generates a valid set of mixture parameters, the partitioned network output

$$\hat{Y} = (\tilde{\pi}_1, ..., \tilde{\pi}_K, \tilde{\boldsymbol{\mu}}_1, ..., \tilde{\boldsymbol{\mu}}_K, \tilde{\boldsymbol{\sigma}}_1, ..., \tilde{\boldsymbol{\sigma}}_K, \tilde{\boldsymbol{\rho}}_1, ..., \tilde{\boldsymbol{\rho}}_K),$$

with

$$\tilde{\pi}_k \in \mathbb{R}, \ \tilde{\boldsymbol{\mu}}_k \in \mathbb{R}^d, \ \tilde{\boldsymbol{\sigma}}_k \in \mathbb{R}^d \text{ and } \tilde{\boldsymbol{\rho}}_k \in \mathbb{R}^{\frac{d^2-d}{2}}$$

is further transformed to meet parameter value requirements, i.e.

$$\pi_k = \text{softmax}(\tilde{\pi}_1, ..., \tilde{\pi}_K)_k,$$

such that $\sum_k \pi_k = 1$ and

$$\boldsymbol{\mu}_k = \tilde{\boldsymbol{\mu}}_k$$
$$\sigma_{k,i} = f_\sigma(\tilde{\sigma}_{k,i}) > 0 \ \forall i \in \{1, ..., d\}$$
$$\rho_{k,j} = f_\rho(\tilde{\rho}_{k,j}) \in [-1, 1] \ \forall j \in \left\{1, ..., \frac{d^2-d}{2}\right\}.$$

Note that the covariance matrices $\boldsymbol{\Sigma}_k$ are calculated from the standard deviations and correlations in order to ensure positive definiteness. Common choices for $f_\sigma$ are given by the *exponential function* (Bishop, 1994), a shifted version of the *Exponential Linear Unit* (Clevert et al., 2015; Guillaumes, 2017) and the *softplus* function (Dugas et al., 2001; Glorot et al., 2011; Iso et al., 2017). For $f_\rho$, an alternative to the originally proposed *tanh* function is given by the *softsign* function (Glorot & Bengio, 2010; Iso et al., 2017).

For building a probabilistic sequence model using MDNs, a common choice is the Sequence MDN model as proposed by Graves (2013). Here, an MDN is stacked on top of an LSTM (Hochreiter & Schmidhuber, 1997) network. The recurrent structure is then used for encoding an input sequence as well as for generating predictions one step at a time.

**The $\mathcal{N}$-MDN variant:** Hug et al. (2020) introduced an MDN variant for probabilistic sequence modeling, which avoids the need for Monte Carlo methods when generating multi-modal distributions over sequences by directly parameterizing mixtures of $\mathcal{N}$-Curves, referred to as $\mathcal{N}$-MDN. The model consists of two main components, a recurrent neural network (an LSTM in this case) used as a sequence encoder and an MDN. During training, the $\mathcal{N}$-MDN learns to map a given input sequence onto an $\mathcal{N}$-Curve (mixture) modeling a distribution over target sequences, e.g. subsequent sequence elements in the case of sequence prediction. To achieve this, the $\mathcal{N}$-MDN's LSTM network encodes the input into a vector representation $\mathbf{v}$, which is then passed into the MDN in order to map it onto the parameters of said $\mathcal{N}$-Curve (mixture), i.e. the mixing weights, as well as the mean vectors and covariance matrices of the Gaussian control points.

## 3 $\mathcal{N}$-Curve Gaussian Processes

With $\mathcal{N}$-Curves providing a representation for stochastic processes $\{X_t\}_{t \in T}$ comprised of Gaussian random variables $X_t \sim \mathcal{N}(\boldsymbol{\mu}, \boldsymbol{\Sigma})$, we first show that $\mathcal{N}$-Curves are a special case of GPs with an implicit mean and covariance function[3]. Following the definition of GPs (MacKay, 2003; Rasmussen & Williams, 2006), an $\mathcal{N}$-Curve can be classified as a GP, if for any finite subset $\{t_1, ..., t_N\}$ of $T$, the joint probability density $p(X_{t_1}, ..., X_{t_N})$ of corresponding random variables is Gaussian. This property is referred to as the *GP property*. We show that this property holds by reformulating the curve construction formula (see Eq. 1) into a linear transformation[4] $\boldsymbol{X} = \mathbf{C} \cdot \boldsymbol{P}$ of the Gaussian control points stacked into a $((L+1) \cdot d \times 1)$ vector

$$\boldsymbol{P}^\top = \begin{pmatrix} P_0^\top & P_1^\top & \cdots & P_L^\top \end{pmatrix} \tag{7}$$

using a $(N \cdot d \times (L+1) \cdot d)$ transformation matrix

$$\mathbf{C} = \begin{pmatrix} \mathbf{B}_{0,L}(t_1) & \dots & \mathbf{B}_{L,L}(t_1) \\ \vdots & \ddots & \vdots \\ \mathbf{B}_{0,L}(t_N) & \dots & \mathbf{B}_{L,L}(t_N) \end{pmatrix} \tag{8}$$

---

[3]The $\mathcal{N}$-Curve representation does not require an explicit kernel or covariance function, as is the common representation for Gaussian processes, because the correlation between points on the curve emerges from the geometric constraints of the underlying Bézier curve. The curve function itself provides the mean function.

[4]For clarity, multivariate random variables may be written in bold font occasionally.

determined by the Bernstein polynomials, with $\mathbf{B}_{l,L}(t_j) = b_{l,L}(t_j)\mathbf{I}_d$ and $t_j \in T_N$. As $\boldsymbol{P}$ is itself a Gaussian random vector, $\boldsymbol{X}$ and its corresponding probability density $p(\boldsymbol{X}) = p(X_1, ..., X_N)$ are also Gaussian.

As the Gaussians along an $\mathcal{N}$-Curve are correlated through the shared set of curve control points, the mean and kernel functions of the induced GP, denoted as $\mathcal{N}$-GP in the following, can be given explicitly. In the following sections, we thus derive the $\mathcal{N}$-GP for the univariate, multivariate and multi-modal case, with respective mean and kernel functions.

### 3.1  Univariate $\mathcal{N}$-Curve Gaussian Processes

We first consider univariate GPs, which target scalar-valued functions $f : \mathbb{R} \to \mathbb{R}$, as these on provide the most common use case for GPs and also grant us a simple case for deriving the mean and kernel functions induced by a given $\mathcal{N}$-Curve while also allowing a visual examination of some properties of the $\mathcal{N}$-GP. Here, the stochastic control points $P_l$ are defined by the mean value $\mu_l$ and variance $\sigma_l^2$. The mean function is equivalent to Eq. 2. Thus, we focus on the kernel $k_{\mathcal{P}}(t_i, t_j)$ for two curve points $X = f(t_i) = \sum_{l=0}^{L} b_{l,L}(t_i)P_l$ and $Y = f(t_j) = \sum_{l=0}^{L} b_{l,L}(t_j)P_l$ at indices $t_i$ and $t_j$ with $t_i, t_j \in [0, 1]$. The respective mean values are given by $\mu_X = \sum_{l=0}^{L} b_{l,L}(t_i)\mu_l$ and $\mu_Y = \sum_{l=0}^{L} b_{l,L}(t_j)\mu_l$. From $k(t_i, t_j) = \mathrm{cov}(f(t_i), f(t_j))$ then follows:

$$
\begin{aligned}
k_{\mathcal{P}}(t_i, t_j) &= \mathbb{E}[(X - \mu_X)(Y - \mu_Y)] \\
&= \mathbb{E}\left[\left(\sum_{l=0}^{L} b_{l,L}(t_i)P_l - \mu_X\right)\left(\sum_{l=0}^{L} b_{l,L}(t_j)P_l - \mu_Y\right)\right] \\
&= \mathbb{E}\left[\sum_{l=0}^{L} b_{l,L}(t_i)b_{l,L}(t_j)P_l^2\right] + \mathbb{E}\left[\sum_{l=0}^{L}\left(\sum_{l'=0, l'\neq l}^{L} b_{l,L}(t_i)b_{l',L}(t_j)P_l P_{l'}\right)\right] \\
&\quad - \mu_Y \underbrace{\sum_{l=0}^{L} b_{l,L}(t_i)\mu_l}_{=\mu_Y\mu_X} - \mu_X \underbrace{\sum_{l=0}^{L} b_{l,L}(t_j)\mu_l}_{=\mu_X\mu_Y} + \mu_X\mu_Y.
\end{aligned}
$$

With $\mathbb{E}[P_i \cdot P_j] = \mathbb{E}[P_i] \cdot \mathbb{E}[P_j]$, which follows from the independence of the control points, and $\mathbb{E}[P_i^2] = \mathrm{Var}[P_i] + (\mathbb{E}[P_i])^2$, we obtain the closed-form solution

$$
k_{\mathcal{P}}(t_i, t_j) = \sum_{l=0}^{L} b_{l,L}(t_i)b_{l,L}(t_j)(\sigma_l^2 + \mu_l^2) + \sum_{l=0}^{L}\left(\sum_{l'=0, l'\neq l}^{L} b_{l,L}(t_i)b_{l',L}(t_j)\mu_l\mu_{l'}\right) - \mu_X\mu_Y, \tag{9}
$$

where $k_{\mathcal{P}}(t_i, t_j)$ is a positive semi-definite function[5] for any $t_* \in [0, 1]$ and arbitrary sets of Gaussian control points with mean values $\mu_l$ and variances $\sigma_l$.

As the $\mathcal{N}$-GP is strongly dependent on the given set of control points, it allows for a range of different kernels. For a brief comparison, Fig. 1 illustrates two standard kernels (Görtler et al., 2019), given by a *radial basis function* (RBF) kernel

$$
k_{\sigma, l}^{\mathrm{rbf}}(t_i, t_j) = \sigma^2 \exp\left(-\frac{\|t_i - t_j\|^2}{2l^2}\right), \tag{10}
$$

with $\sigma = 1$ and $l = 0.25$ and a linear kernel

$$
k_{\sigma, \sigma_b, c}^{\mathrm{lin}}(t_i, t_j) = \sigma_b^2 + \sigma^2(t_i - c)(t_j - c), \tag{11}
$$

with $\sigma = \sigma_b = c = 0.5$, and two $\mathcal{N}$-GP kernels $k_{\mathcal{P}_1}(t_i, t_j)$ and $k_{\mathcal{P}_2}(t_i, t_j)$. $\mathcal{P}_1$ consists of two unit Gaussians and $\mathcal{P}_2$ consists of 9 zero mean Gaussian control points with standard deviations $\sigma_0 = \sigma_8 = 1$, $\sigma_1 = \sigma_7 = 1.25$, $\sigma_2 = \sigma_6 = 1.5$, $\sigma_3 = \sigma_5 = 1.75$ and $\sigma_4 = 2$. The standard deviations vary in order to cope with non-linear blending (see Eq. 3). The Gram matrices calculated from 20 equally spaced values in $[0, 1]$ are depicted for each kernel.

---

[5]A proof for the positive semi-definiteness of the $\mathcal{N}$-GP kernel is provided in appendix A.

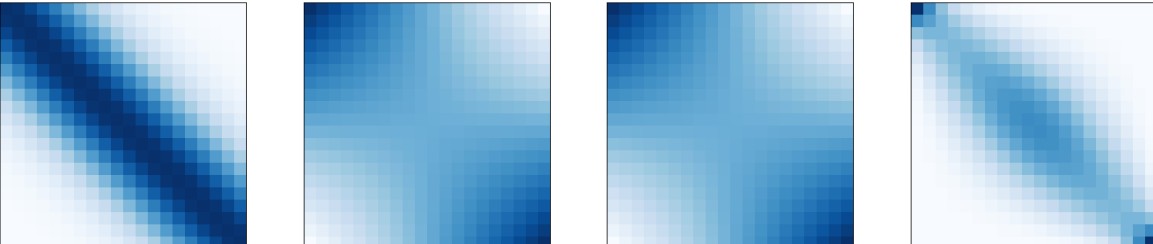

Figure 1: Gram matrices for 20 equally spaced values in $[0, 1]$ obtained by using different GP kernels, showing parallels between different $\mathcal{N}$-GP kernels and the RBF and linear kernels. Left to right: RBF kernel, linear kernel and $\mathcal{N}$-GP kernels $k_{\mathcal{P}_1}(t_i, t_j)$ and $k_{\mathcal{P}_2}(t_i, t_j)$.

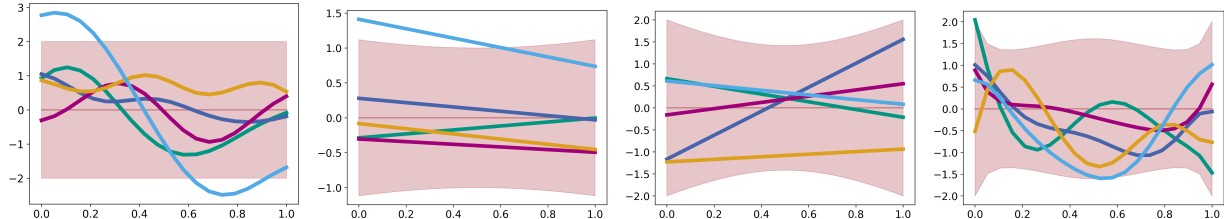

Figure 2: Samples drawn from prior distributions using different GP kernels. The $2\sigma$ region is depicted as a red shaded area. Left to right: RBF kernel, linear kernel and $\mathcal{N}$-GP kernels $k_{\mathcal{P}_1}(t_i, t_j)$ and $k_{\mathcal{P}_2}(t_i, t_j)$.

When comparing the Gram matrices, it can be seen, that the matrix calculated with $k_{\mathcal{P}_1}$ is equal to that calculated with $k^{\mathrm{lin}}_{\sigma = \sigma_b = c = 0.5}(t_i, t_j)$ when normalizing its values to $[0, 1]$. On the other hand, the matrix obtained with $k_{\mathcal{P}_2}$, which is derived from a more complex $\mathcal{N}$-Curve, tends to be more comparable to the matrix calculated with $k^{\mathrm{rbf}}_{\sigma = 1, l = 0.25}(t_i, t_j)$. These parallels are also visible when comparing sample functions drawn from each GP prior, assuming a zero mean GP using the different kernels as depicted in Fig. 2.

As a final note, the $\mathcal{N}$-GP is non-stationary, as its kernel depends on the actual values of $t_i$ and $t_j$. Further, it is non-periodic and its smoothness is controlled by the underlying Bézier curve, i.e. the position and number of control points.

## 3.2 Multivariate $\mathcal{N}$-Curve Gaussian Processes

Multivariate GPs target vector-valued functions $\mathbf{f}(t)$, which map scalar inputs onto $d$-dimensional vectors, e.g. $\mathbf{f} : \mathbb{R} \rightarrow \mathbb{R}^d$. Following this, for elevating our univariate $\mathcal{N}$-GP derived in the previous section to the multivariate case, there exist two closely related approaches we can adopt. The first sticks with the multivariate Gaussian distribution and models matrix-valued random variables by using stacked mean vectors in combination with block partitioned covariance matrices (Álvarez et al., 2012). The other revolves around the matrix normal distribution (Chen et al., 2020a;b), which can be transformed into a multivariate Gaussian distribution by vectorizing the mean matrix and calculating the covariance matrix as the Kronecker product of both scale matrices, thus establishing a connection to the former.

We adopt the first approach, as it simplifies the extension of the univariate $\mathcal{N}$-GP. Following this, the Gram matrix of a $d$-variate GP for a finite index subset with $|T_N| = N$ is defined by the $(Nd \times Nd)$ block partitioned matrix

$$\Sigma = \begin{pmatrix} \mathbf{K}_{\mathcal{P}}(t_1, t_1) & \cdots & \mathbf{K}_{\mathcal{P}}(t_1, t_N) \\ \vdots & \ddots & \vdots \\ \mathbf{K}_{\mathcal{P}}(t_N, t_1) & \cdots & \mathbf{K}_{\mathcal{P}}(t_N, t_N) \end{pmatrix} \tag{12}$$

calculated using the matrix-valued kernel $\mathbf{K}_{\mathcal{P}}(t_i, t_j) = \text{cov}(\boldsymbol{X}, \boldsymbol{Y})$. Here, $\boldsymbol{X} = \sum_{l=0}^{L} b_{l,L}(t_i)\boldsymbol{P}_l$ and $\boldsymbol{Y} = \sum_{l=0}^{L} b_{l,L}(t_j)\boldsymbol{P}_l$ are now $d$-variate Gaussian random variables resulting from the linear combination of $d$-variate $\mathcal{N}$-Curve control points $\boldsymbol{P}_l$. Thus, the multivariate generalization of Eq. 9 yields a $(d \times d)$ matrix and is given by

$$
\begin{aligned}
\mathbf{K}_{\mathcal{P}}(t_i, t_j) &= \mathbb{E}[(\boldsymbol{X} - \boldsymbol{\mu}_X)(\boldsymbol{Y} - \boldsymbol{\mu}_Y)^{\top}] \\
&= \mathbb{E}\left[\boldsymbol{X}\boldsymbol{Y}^{\top}\right] - \underbrace{\mathbb{E}\left[\boldsymbol{X}\boldsymbol{\mu}_Y^{\top}\right]}_{=\boldsymbol{\mu}_X\boldsymbol{\mu}_Y^{\top}} - \underbrace{\mathbb{E}\left[\boldsymbol{\mu}_X\boldsymbol{Y}^{\top}\right]}_{=\boldsymbol{\mu}_X\boldsymbol{\mu}_Y^{\top}} + \boldsymbol{\mu}_X\boldsymbol{\mu}_Y^{\top} \\
&= \sum_{l=0}^{L} b_{l,L}(t_i)b_{l,L}(t_j)\left(\boldsymbol{\Sigma}_l + \boldsymbol{\mu}_l\boldsymbol{\mu}_l^{\top}\right) + \sum_{l=0}^{L}\left(\sum_{l'=0, l'\neq l}^{L} b_{l,L}(t_i)b_{l',L}(t_j)\boldsymbol{\mu}_l\boldsymbol{\mu}_{l'}^{\top}\right) - \boldsymbol{\mu}_X\boldsymbol{\mu}_Y^{\top}.
\end{aligned}
\tag{13}
$$

The $(n \times d)$ mean vector is defined as the concatenation of all mean vectors $\boldsymbol{\mu}_{\mathcal{P}}(t_i)$ (see also Eq. 2) along the underlying Bézier curve, i.e.

$$
(\mathbf{m}_{\mathcal{P}}(T_N))^{\top} = \left((\boldsymbol{\mu}_{\mathcal{P}}(t_1))^{\top} \quad \cdots \quad (\boldsymbol{\mu}_{\mathcal{P}}(t_N))^{\top}\right).
\tag{14}
$$

### 3.3 Multi-modal $\mathcal{N}$-Curve Gaussian Processes

With sequence modeling tasks often being multi-modal problems and GPs as presented before being incapable of modeling such data, we consider multi-modal GPs as a final case. A common approach for increasing the expressiveness of a statistical model, e.g. for heteroscedasticity or multi-modality, is to employ a mixture model. Thereby, rather than a single model or distribution, a mixture of which are used with each component in the mixture covering a subset of the data. A widely used mixture model is given by the Gaussian mixture model (Bishop, 2006), which is defined as a convex combination of $K$ Gaussian distributions with (mixing) weights $\boldsymbol{\pi} = \{\pi_1, ..., \pi_K\}$ and probability density function

$$
p(\mathbf{x}) = \sum_{k=1}^{K} \underbrace{p(z = k)}_{\pi_k}\underbrace{p(\mathbf{x}|z = k)}_{\mathcal{N}(\boldsymbol{\mu}_k, \boldsymbol{\Sigma}_k)}, \; z \sim \text{Categorical}(\boldsymbol{\pi}).
\tag{15}
$$

Transferred to GPs, a popular approach is given by the *mixture of Gaussian process experts* (Tresp, 2000; Rasmussen & Ghahramani, 2001; Yuan & Neubauer, 2008), which extends on the mixture of experts model (Jacobs et al., 1991). In this approach, the mixture model is defined in terms of $K$ GP *experts* $\mathcal{G}_k$ with mean function $\mathbf{m}_k$ and kernel $\mathbf{K}_k$

$$
\sum_{k=1}^{K} p(z = k|\mathbf{x})\mathcal{G}_k(\mathbf{m}_k(\cdot), \mathbf{K}_k(\cdot, \cdot)),
\tag{16}
$$

using a conditional weight distribution $\text{Categorical}(\boldsymbol{\pi}|\mathbf{x})$ for a given sample $\mathbf{x}$. The weight distribution is generated by a *gating network*, which decides on the influence of each local *expert* for modeling a given sample. This is the key difference to the Gaussian mixture model, where the weight distribution is static and determined a priori (e.g. via EM (Dempster et al., 1977) or an MDN (Bishop, 1994)). It can be noted that mixtures of experts are also often used to lower the computational load of a GP model, as less data points have to be considered during inference due to the use of local experts (e.g. Deisenroth & Ng (2015); Lederer et al. (2021)).

In line with the approach described by Hug et al. (2020), which builds on Gaussian mixture models, we define the multi-modal extension of our $\mathcal{N}$-GP as a mixture of $K$ $\mathcal{N}$-GPs

$$
\mathcal{MG}(\boldsymbol{\pi}, \{\mathcal{G}_k\}_{k \in \{1,...,K\}}) = \sum_{k=1}^{K} \pi_k \mathcal{G}_k(\mathbf{m}_{\mathcal{P}_k}(\cdot), \mathbf{K}_{\mathcal{P}_k}(\cdot, \cdot)),
\tag{17}
$$

with $\mathcal{N}$-GP components $\mathcal{G}_k$ and the static prior weight distribution $\boldsymbol{\pi} = \{\pi_1, ...\pi_K\}$ with $\sum_{k=1}^{K} \pi_k = 1$. The mean and kernel functions are determined separately for each GP component according to equations 2 and 9 (unimodal case), or 13 and 14 (multi-modal case). Given these functions and the weights $\pi$, the mixture distribution can be evaluated at a given index.

## 4 Bayesian Inference for $\mathcal{N}$-Curve Mixture Density Networks

In order to enable GP-based Bayesian inference on top of the regression-based $\mathcal{N}$-MDN model, we propose to use the $\mathcal{N}$-MDN as a data-dependent generator for prior distributions within the GP framework, where the $\mathcal{N}$-MDN is adapted to a given dataset through training[6]. Given an input sequence, we achieve this by calculating an $\mathcal{N}$-GP's mean and kernel functions from each corresponding $\mathcal{N}$-Curve output by the $\mathcal{N}$-MDN, yielding a prior for a GP modeling a respective target sequence. Despite the $\mathcal{N}$-MDN being a competetive[7] probabilistic sequence model by itself, the combination of both approaches yields mutual advantages by providing a convenient and stable way of estimating the $\mathcal{N}$-GP's parameters and enhancing the $\mathcal{N}$-MDN's flexibility and expressiveness in an post-hoc manner by creating the possibility of calculating different posterior predictive distributions from the $\mathcal{N}$-MDN's output in light of (new) observations via the induced $\mathcal{N}$-GP.

For conciseness, in the following sections we focus on the task of (probabilistic) sequence prediction, where given a length $N_{\text{in}}$ input sequence (*observation*), the subsequent $N_{\text{pred}}$ sequence elements need to be predicted. The length of each sequence is referred to as the observable and predictive time horizon, respectively. First, we discuss possible real-world use cases of the combined approach (section 4.1). Afterwards, we briefly summarize how the $\mathcal{N}$-MDN network is trained (section 4.2) and how posterior predictive distributions corresponding to the discussed use cases are calculated using our proposed combination of $\mathcal{N}$-MDNs and $\mathcal{N}$-GPs (section 4.3).

### 4.1 Practical Implications and Use cases

To put the value of our combined model into perspective from a practical standpoint, we discuss different use cases that we expect to benefit from the $\mathcal{N}$-GP extension, namely prediction *refinement* and *update*.

**Refinement:** We first examine the $\mathcal{N}$-GP as a tool for improving the overall prediction performance considering different posterior predictive distributions given different subsets of the input sequence. More formally, given an input sequence $\{\mathbf{x}_1, ..., \mathbf{x}_{N_{\text{in}}}\}$, which is processed by the $\mathcal{N}$-MDN yielding the $\mathcal{N}$-GP's prior distribution $p(X_1, ..., X_{N_{\text{in}}+N_{\text{pred}}})$, a refinement of this initial prediction is given by a posterior predictive distribution

$$p(\{X_1, ..., X_{N_{\text{in}}+N_{\text{pred}}}\} \backslash \{X_i\}_{i \in \mathcal{I}} | \{\mathbf{x}_i\}_{i \in \mathcal{I}}) \tag{18}$$

with $\mathcal{I} \subseteq \{1, ..., N_{\text{in}}\}$ giving a subset of the input sequence. We expect this refinement to improve the prediction performance, as the original maximum likelihood prediction generated by the $\mathcal{N}$-MDN tends to average out small variations in the data and the refinement procedure adapts the prediction more towards the actual observation in a controlled way. We consider this the most practically relevant use case, as it directly affects the model's accuracy.

**Update:** Another interesting option opened up by embedding the $\mathcal{N}$-MDN into the GP framework is given by the use case of updating a multi-step prediction under the presence of new data within the predictive time horizon. For this, we calculate a posterior predictive distribution similar to Eq. 18 (refinement), but condition on a new observation within the predictive time horizon instead, i.e. $\mathcal{I} \subseteq \{N_{\text{in}}+1, ..., N_{\text{in}}+N_{\text{pred}}\}$. As sequence models usually predict several time steps into the future, an easy to calculate and fast to compute update to the prediction under the presence of new data can be valuable. The $\mathcal{N}$-GP enables such updates without the need for additional passes through the underlying neural network. Further, it is

---

[6]Training the $\mathcal{N}$-MDN does not involve the $\mathcal{N}$-GP component.

[7]A comparison with other probabilistic sequence models commonly used in the context of human trajectory prediction is given in appendix B.

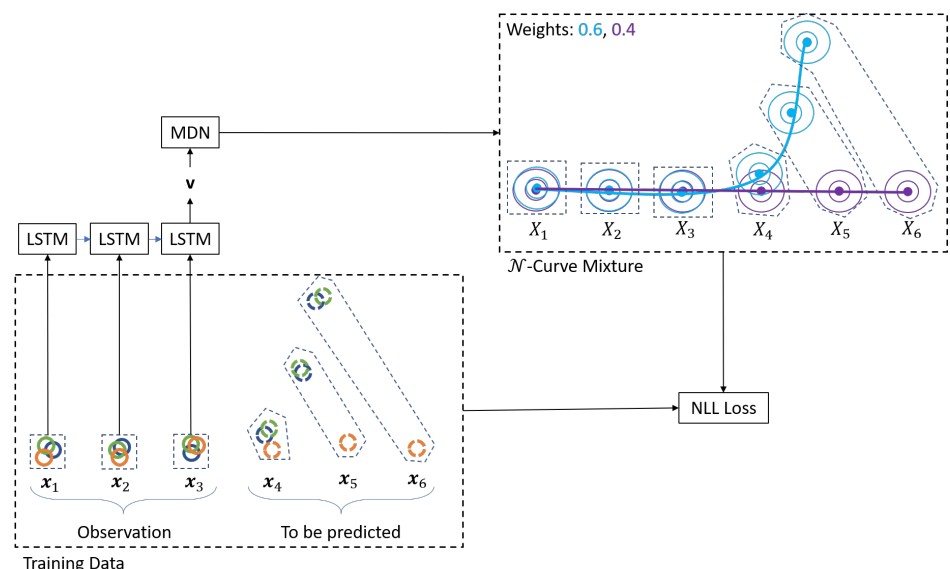

Figure 3: Schematic of $\mathcal{N}$-MDN training for sequences of length $N = 6$, with $N_{\mathrm{in}} = 3$ and $N_{\mathrm{pred}} = 3$.

unaffected by potentially missing intermediate observations. This is especially valuable, as common sequence prediction models require complete sequences as input, where such gaps need to be filled with data extracted from the model's own prediction. This can be problematic under the presence of multiple modes in the predicted distribution, making Monte Carlo methods a necessity. In contrast, within the GP framework missing information between observed data points is naturally interpolated. Following this, when certain requirements are met, our model allows to fill gaps in light of fragmented observations easily. First, a full $N_{\mathrm{in}}$-step input sequence is required for the underlying $\mathcal{N}$-MDN for generating our prior distribution. Second, the gaps to fill must be within the modeled predictive time horizon.

## 4.2 Parameter Estimation

In our approach, the $\mathcal{N}$-GP relies on prior distributions generated by an $\mathcal{N}$-MDN, whose parameters are learned from data. Following Hug et al. (2020), the $\mathcal{N}$-MDN is trained using a set of $M$ fixed-length trajectories $\mathcal{D} = \{\mathcal{X}_1, ..., \mathcal{X}_M\}$ with $\mathcal{X}_i = \{\mathbf{x}_1^i, ..., \mathbf{x}_N^i\}$, applying the negative log-likelihood loss

$$\mathcal{L} = \frac{1}{M} \sum_{j=1}^{M} - \log \sum_{k=1}^{K} \exp \left\{ \log \pi_k + \sum_{i=1}^{N} \log \left( \mathcal{N}(\mathbf{x}_i^j | \mu_{\mathcal{P}}(t_i), \Sigma_{\mathcal{P}}(t_i)) \right) \right\} \tag{19}$$

in conjunction with a gradient descent policy.

In order to give an illustrative example considering a sequence prediction task, training sequences of length $N$ are split into an observed portion of length $N_{\mathrm{in}}$, which is used as input for the model, and a to be predicted portion of length $N_{\mathrm{pred}}$, where $N = N_{\mathrm{in}} + N_{\mathrm{pred}}$. The $\mathcal{N}$-MDN then outputs an $\mathcal{N}$-Curve (mixture), which models full sequences of length $N$. This ensures that within the GP framework, we are able to condition on elements within the input sequence. The negative log-likelihood loss function is calculated given full training sequences. A schematic of this is depicted in Fig. 3.

## 4.3 Conditional Inference

Finally, we want to give a brief overview of how (conditional) inference works within our combined model, especially with regard to the refinement and update use cases discussed in section 4.1. Given an input sequence, the $\mathcal{N}$-MDN generates an $\mathcal{N}$-Curve mixture, which models the input as well as possible future sequences. Using Equations 13, 14 and 17, we calculate the $\mathcal{N}$-GP prior $p(X_1, ..., X_N)$ from this mixture,

which is a joint $K$-component Gaussian mixture distribution over all $N$ modeled time steps. Now, for determining a conditional posterior predictive distribution, we first partition the joint prior distribution into a partition containing the $N_{\mathrm{in}}$ time steps to condition on and the remaining $N_{\mathrm{pred}}$ time steps

$$
\begin{aligned}
p(X_1, ..., X_N) &= p(\{X_1, ..., X_{N_{\mathrm{in}}}\} \cup \{X_{N_{\mathrm{in}}+1}, ..., X_N\}) \\
&= p(X_A \cup X_B),
\end{aligned}
\tag{20}
$$

with component mean vectors and covariance matrices partitioned as

$$
\boldsymbol{\mu}_k = \begin{bmatrix} \boldsymbol{\mu}_{kA} \\ \boldsymbol{\mu}_{kB} \end{bmatrix}, \quad \boldsymbol{\Sigma}_k = \begin{bmatrix} \boldsymbol{\Sigma}_{kAA} & \boldsymbol{\Sigma}_{kAB} \\ \boldsymbol{\Sigma}_{kBA} & \boldsymbol{\Sigma}_{kBB} \end{bmatrix}.
$$

Following e.g. (Bishop, 2006; Petersen & Pedersen, 2008; Murphy, 2022), the posterior weights, mean vectors and covariance matrices, conditioned on $\mathbf{x}_B$, can then be directly calculated as

$$
\begin{aligned}
\pi_{kA|B} &= \frac{\pi_k \mathcal{N}(\mathbf{x}_B | \boldsymbol{\mu}_{kB}, \boldsymbol{\Sigma}_{kBB})}{\sum_j \pi_j \mathcal{N}(\mathbf{x}_B | \boldsymbol{\mu}_{jB}, \boldsymbol{\Sigma}_{jBB})} \\
\boldsymbol{\mu}_{kA|B} &= \boldsymbol{\mu}_{kA} + \boldsymbol{\Sigma}_{kAB} \boldsymbol{\Sigma}_{kBB}^{-1} (\mathbf{x}_B - \boldsymbol{\mu}_{kB}) \\
\boldsymbol{\Sigma}_{kA|B} &= \boldsymbol{\Sigma}_{kAA} - \boldsymbol{\Sigma}_{kAB} \boldsymbol{\Sigma}_{kBB}^{-1} \boldsymbol{\Sigma}_{kBA}.
\end{aligned}
\tag{21}
$$

The probability distribution for individual trajectory points at time $t$ can be extracted through marginalization using $X_A = \{X_t\}$, e.g.

$$
p(X_A) = \sum_k \pi_k p_k(X_A) = \sum_k \pi_k \mathcal{N}(x_A | \mu_{kAA}, \Sigma_{kAA}).
$$

A schematic of the inference scheme for the refinement and update use cases is provided in Fig. 4. In this example, we assume the $\mathcal{N}$-GP to model 6-element sequences with prior distribution $p(X_1, X_2, X_3, X_4, X_5, X_6)$, where $p(\cdot)$ was calculated from the $\mathcal{N}$-Curve (mixture) output by an $\mathcal{N}$-MDN given an input sequence $\{\mathbf{x}_1, \mathbf{x}_2, \mathbf{x}_3\}$. As we are now within the GP framework, we can calculate different posterior predictive distributions using observed values from the input sequence (refinement use case) or values observed after the initial input (update use case). Exemplary posterior distributions can then be given by $p(X_1, X_2, X_4, X_5, X_6 | \mathbf{x}_3)$ or $p(X_1, X_2, X_4, X_6 | \mathbf{x}_3, \mathbf{x}_5)$ for the refinement and update use cases, respectively.

## 5  Related Work

Finally, we want to summarize our contributions over Hug et al. (2020) and then give a brief overview of additional related areas of research our work tangents, complementing the references given throughout section 3.

**Comparison with (Hug et al., 2020):**  The main contributions provided by Hug et al. (2020) are given by the concept of $\mathcal{N}$-Curves as a representation of a probability distribution over sequences and the $\mathcal{N}$-MDN neural network as a probabilistic sequence model using those curves as their output. In this paper, we take the concept of $\mathcal{N}$-Curves and extend the theory surrounding them by establishing a link to Gaussian processes, providing a proof of equivalence, and deriving the mean and kernel functions for a GP induced by a given $\mathcal{N}$-Curve (the $\mathcal{N}$-GP) for the univariate, multivariate and multi-modal cases. We further extend their sequence prediction model by re-purposing the $\mathcal{N}$-MDN as a data-dependent generator for $\mathcal{N}$-GP prior distributions, with the goal of enhancing the $\mathcal{N}$-MDN in terms of flexibility and expressiveness.

**Multivariate GPs:**  A range of multivariate GPs (also referred to as multi-output GPs) are summarized under the *Mixing Model Hierarchy* (Bruinsma et al., 2020), which classifies GP models by low-rank covariance structure. These GPs are specializations of a *Linear Mixing Model*, which is defined in terms of a diagonal multi-output kernel and a mixing matrix. Apart from these GPs, models such as (Salimbeni & Deisenroth,

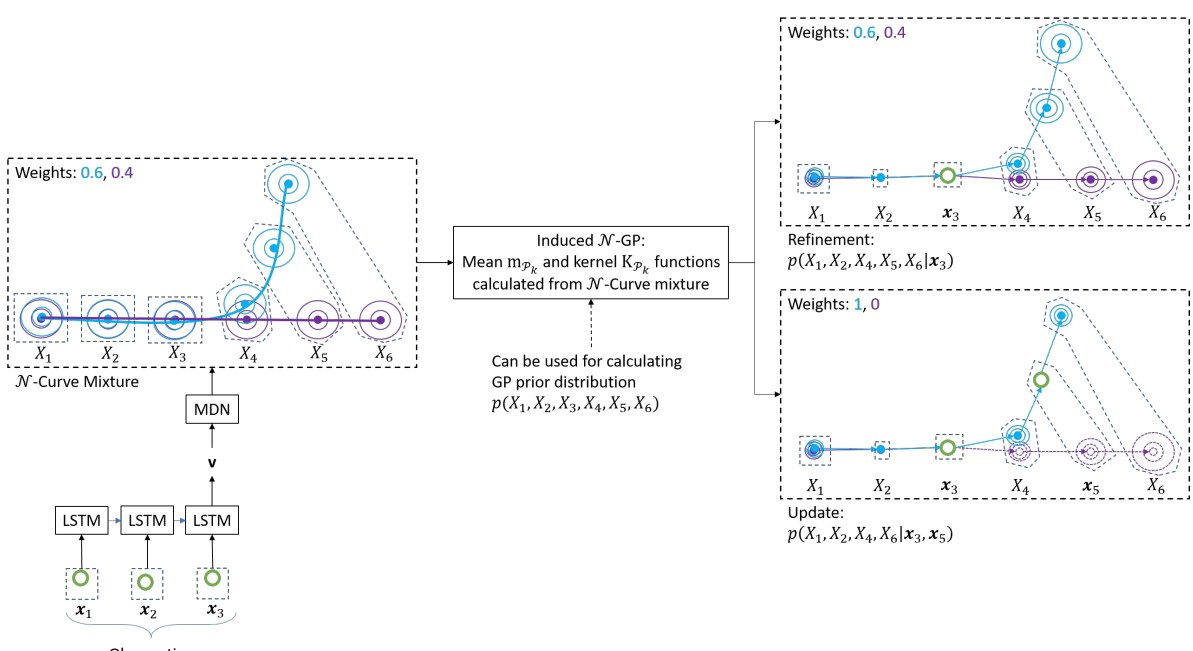

Figure 4: Schematic of the interplay between the $\mathcal{N}$-MDN and $\mathcal{N}$-GP components with regard to the refinement and update use cases. Green circles indicate trajectory points the predictive posterior is conditioned on.

2017) build on deep GPs (Damianou & Lawrence, 2013), which are hierarchies of multiple GPs with non-linear mappings between layers. These approaches commonly decompose the multi-output GP problem into a set of multiple independent single output GP problems via diagonal multivariate kernels. In contrast, in our approach the output dimensionality is tied to the $\mathcal{N}$-Curve control point's dimensionality. Eq. 13 then yields a (non-diagonal) multivariate kernel, where correlations between dimensions are constrained by the shape of the underlying Bézier curve.

**Learning data-dependent priors:** Different approaches that combine kernel methods with deep learning have been proposed, such as *Neural Kernel Networks* (Sun et al., 2018) and *Deep Kernel Learning* (Wilson et al., 2016; Ober et al., 2021). Neural Kernel Networks learn a kernel from data as a composition of primitive kernels, whereas Deep Kernel Learning uses a neural network to map inputs into an intermediate feature space, which then serves as the input space for a GP. Opposed to that, we are using a neural network for mapping inputs onto the parameters of an $\mathcal{N}$-Curve (mixture), from which conditional prior distributions can be derived directly.

**Bernstein basis:** The Bernstein polynomials have been used previously for approximating prior distributions. Petrone (1999b) proposes the use of Bernstein polynomials for approximating a dirichlet process to define prior distributions and MCMC based posterior approximation. These priors are then for example applied for Bayesian density estimation (Petrone, 1999a). Besides that, Jørgensen & Osborne (2022) proposed an approach building on the same core idea[8] of deriving a GP from probabilistic Bézier curves, or Bézier surfaces in their case. However, their work puts strong emphasis on modeling spatial input, while our approach focuses on probabilistic (multi-modal) sequence modeling. Apart from that, one of their main concerns is given by scalability, which is a frequent issue with Gaussian processes. We, on the other hand, rather focus on the derivation of mean and kernel functions for different use-cases. Thereby, we also provide an equivalence proof for $\mathcal{N}$-Curves being Gaussian processes, explore some properties of the $\mathcal{N}$-GP and show surface-level comparisons to other Gaussian process kernels. Finally, on a methodological level, they employ

---

[8]Approach developed concurrently to ours.

an approach based on variational inference for parameter estimation, while we are relying on regression-based MDNs, thus providing an alternative learning strategy.

# 6   Evaluation

In this section we aim to support our proposed combination of $\mathcal{N}$-MDNs with $\mathcal{N}$-GPs in its capabilities of enhancing the $\mathcal{N}$-MDN's flexibility, expressiveness and overall performance, considering the *refinement* and *update* use cases in the scope of an established sequence processing task, i.e. (human) trajectory prediction, using standard benchmarking datasets. This task provides easy to visualize results while also providing sufficient of complexity being a highly multi-modal problem, despite its low data dimensionality. In trajectory prediction, given $N_{\text{in}}$ points of a trajectory as input, a sequence model is tasked to predict the subsequent $N_{\text{pred}}$ trajectory points. Here, the $\mathcal{N}$-MDN models both, the observed and to be predicted trajectory, in order to enable GP-based inference. It should be noted, that we omit a state-of-the-art comparison of the base $\mathcal{N}$-MDN. The reasons for this are two-fold. On the one hand, the model has been proven viable and competetive on the human trajectory prediction task (Hug et al., 2020). On the other hand, this allows us to keep the evaluation focused on our combined model and the use cases discussed in section 4.1. For these reasons, we refer the reader to appendix B for additional comparisons of the base $\mathcal{N}$-MDN with other probabilistic neural sequence models relevant in the context of human trajectory prediction.

## 6.1   Experimental Setup

For the evaluation, we consider scenes from commonly used datasets: *BIWI Walking Pedestrians* (Pellegrini et al. (2009), scenes: ETH and Hotel), *Crowds by Example* (Lerner et al. (2007), scenes: Zara1 and Zara2) and the *Stanford Drone Dataset* (Robicquet et al. (2016), scenes: Bookstore and Hyang). Following common practice, the annotation rate of each dataset is adjusted to 2.5 annotations per second. Further, the evaluation is conducted on trajectories of fixed length $N = N_{\text{in}} + N_{\text{pred}} = 8 + 12$. We trained one $\mathcal{N}$-MDN independently for each dataset to generate $\mathcal{N}$-Curve mixtures, which model complete trajectories of length $N = 20$, given the first $N_{\text{in}} = 8$ trajectory points as input to the model. For training, 80% of each dataset are used.

For a quantitative and qualitative evaluation, we are first considering the refinement use case. For each test dataset, the respective $\mathcal{N}$-MDN is used to generate an $\mathcal{N}$-Curve mixture prediction for every test trajectory, yielding initial predictions $p(X_1, ..., X_N)$, i.e. the $\mathcal{N}$-GP prior. These initial predictions are then refined by calculating the posterior distributions

$$p(\{X_1, ..., X_N\} \backslash \{X_{N_{\text{in}}}\} | \mathbf{x}_{N_{\text{in}}})$$

and

$$p(\{X_1, ..., X_N\} \backslash \{X_4, X_{N_{\text{in}}}\} | \mathbf{x}_4, \mathbf{x}_{N_{\text{in}}}),$$

conditioning on the input's last point $\mathbf{x}_{N_{\text{in}}}$ (*refinement A*) and on $\{\mathbf{x}_4, \mathbf{x}_{N_{\text{in}}}\}$ (*refinement B*), respectively. For more details on how the posterior distributions are obtained, we want to refer the reader to section 4.3. By increasing the number of observed points, we expect the prediction to adapt towards a given sample trajectory. For measuring the performance, we apply the *Average Displacement Error* (ADE, Kothari et al. (2021)) according to the standard evaluation approach, using a maximum likelihood estimate[9]. As the ADE does not provide an adequate measure for assessing the quality of (multi-modal) probabilistic predictions, we use the *Negative Log-Likelihood* (NLL) in addition to the ADE, which is a common choice for this purpose (Bhattacharyya et al., 2018; Ivanovic & Pavone, 2019). More detailed information on the performance measures is provided in appendix B.2. Both the ADE and the NLL are calculated for the estimate of the observed trajectory, as well as for the time steps to be predicted. For example for refinement B, the marginal distributions

$$p_{\text{obs}}(\{X_1, ..., X_{N_{\text{in}}}\} \backslash \{X_4, X_{N_{\text{in}}}\}) = \int p(\{X_1, ..., X_N\} \backslash \{X_4, X_{N_{\text{in}}}\} | \mathbf{x}_4, \mathbf{x}_{N_{\text{in}}}) d\{X_{N_{\text{in}}+1}, ..., X_N\}$$

---

[9]We are using the mean vectors of the marginalized posterior Gaussian mixture distributions for each time step to be considered.

|  |  | Initial Prediction | Refinement A | Refinement B |
|---|---|---|---|---|
| ETH | ADE | 3.85 / 11.25 | 3.95 / 10.12 | 2.39 / 10.18 |
|  | NLL | 6.51 / 7.58 | 5.43 / 7.08 | -115.09 / 1.70 |
| Hotel | ADE | 5.69 / 17.96 | 4.19 / 17.07 | 2.70 / 16.73 |
|  | NLL | 6.99 / 8.20 | 5.56 / 7.71 | 10.63 / 8.59 |
| Zara1 | ADE | 4.09 / 19.10 | 2.89 / 17.52 | 1.63 / 17.64 |
|  | NLL | 6.83 / 8.18 | 5.27 / 7.63 | -51.93 / 8.15 |
| Zara2 | ADE | 2.98 / 21.38 | 2.64 / 20.07 | 1.69 / 20.05 |
|  | NLL | 6.59 / 8.09 | 5.08 / 7.59 | -60.95 / -1.76 |
| Bookstore | ADE | 4.04 / 17.21 | 3.65 / 15.97 | 2.16 / 16.29 |
|  | NLL | 7.46 / 8.37 | 5.88 / 7.76 | -11.89 / 7.63 |
| Hyang | ADE | 5.51 / 36.46 | 5.01 / 34.05 | 3.16 / 32.18 |
|  | NLL | 8.21 / 9.42 | 6.65 / 8.86 | -49.30 / 9.05 |

Table 1: Quantitative results of the initial (prior) prediction as generated by an $\mathcal{N}$-MDN and posterior refinements. Table entries report the estimation error with respect to the input trajectory ($p_{\text{obs}}$, first value) and the trajectory to be predicted ($p_{\text{pred}}$, second value), respectively. ADE errors are reported in pixels. Lower is better for both performance measures.

and

$$p_{\text{pred}}(X_{N_{\text{in}}+1},...,X_N) = \int p(\{X_1,...,X_N\}\setminus\{X_4,X_{N_{\text{in}}}\}|\mathbf{x}_4,\mathbf{x}_{N_{\text{in}}})d_{\{}X_1,...,X_{N_{\text{in}}}\}\setminus\{X_4,X_{N_{\text{in}}}\}$$

are evaluated. As these posteriors are Gaussian mixture distributions, these marginals can analytically be calculated (see section 4.3). Beyond the refinement use case, we are providing a qualitative example for the update use case. Here, the posterior distribution $p(\{X_1,...,X_N\}\setminus\{X_{N_{\text{in}},X_{14}}\}|\mathbf{x}_{N_{\text{in}}},\mathbf{x}_{14})$ is considered.

## 6.2 Results

The quantitative results for the $\mathcal{N}$-GP – based prediction refinement with respect to the selected performance measures are depicted in Table 1. Overall, an increase in performance can be observed when refining the estimate generated by the $\mathcal{N}$-MDN using 1 and 2 points, respectively. This supports our expectation of an increase in prediction performance through adding GP-based Bayesian inference to the $\mathcal{N}$-MDN. Despite this, however, the results indicate an inconsistent gain in performance comparing both posteriors. Following this, we provide qualitative examples to further investigate these findings.

Two examples highlighting common cases for a positive effect of the refinement on the prediction performance is given in Fig. 5. On the one hand, the refinement can lead to the estimate being pulled closer to the ground truth in the input portion, which expands far into the future prediction. On the other hand, the refinement can lead to the suppression of inadequate mixture components, that were assigned high weights in the prior distribution.

In some cases it can be observed, that conditioning on 2 points (refinement B) degrades the performance in comparison to using a single point (refinement A). With respect to the NLL, this can be attributed to an increased number of trajectory point variances decreasing or even collapsing. Then, even minor inaccuracies in the mean prediction result in higher NLL values, even if the estimate is closer to the ground truth. Looking at the ADE, the loss in performance can most likely be attributed to the enforced interpolation of the condition points, which sometimes leads to unwanted deformations of the mean prediction. One of the main causes for this is given by the input trajectories commonly being subject to noise. Examples for both of these cases are depicted in Fig. 6. It could be noted, that a common approach for dealing with such problems is given by adding an error term to each observed point (Görtler et al., 2019).

Lastly, we briefly showcase our approach considering the prediction update use case. An example for the posterior distribution given an additional observation within the predictive time horizon is depicted in Fig. 7. While there are initially multiple relevant mixture components (according to their weights), the additional

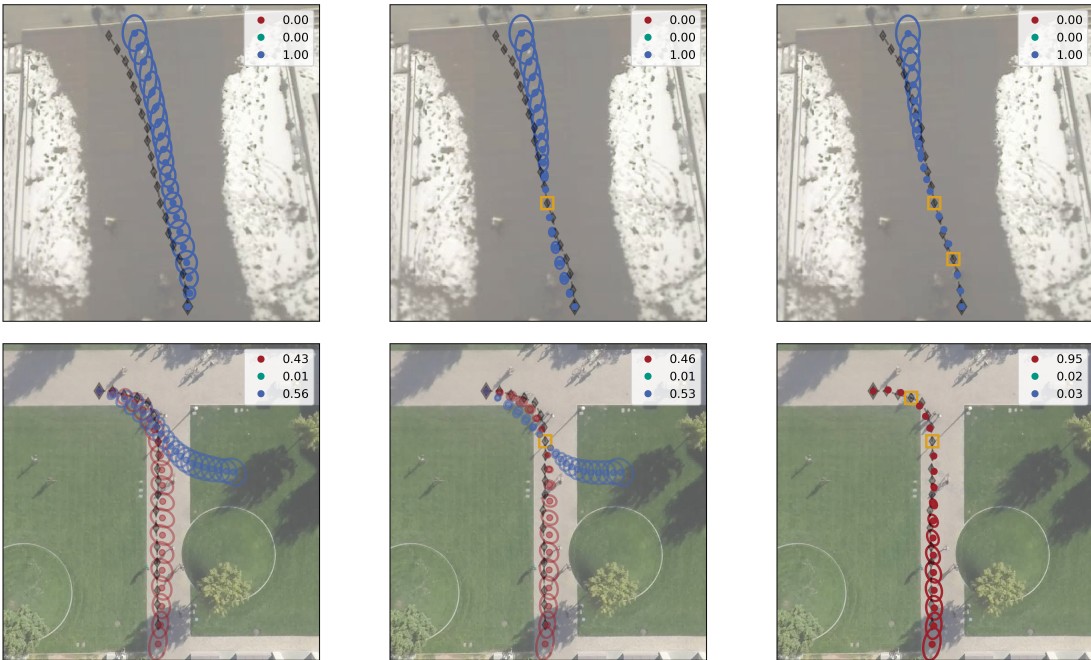

Figure 5: Examples for improved prediction through conditioning on 1 or 2 points, respectively. Left to right: Initial (prior) prediction, refinement A and refinement B. Yellow squares indicate condition points. Ground truth trajectories are depicted in black and start at the bottom (top row) and in the top left (bottom row). The first 8 trajectory points are input into the $\mathcal{N}$-MDN.

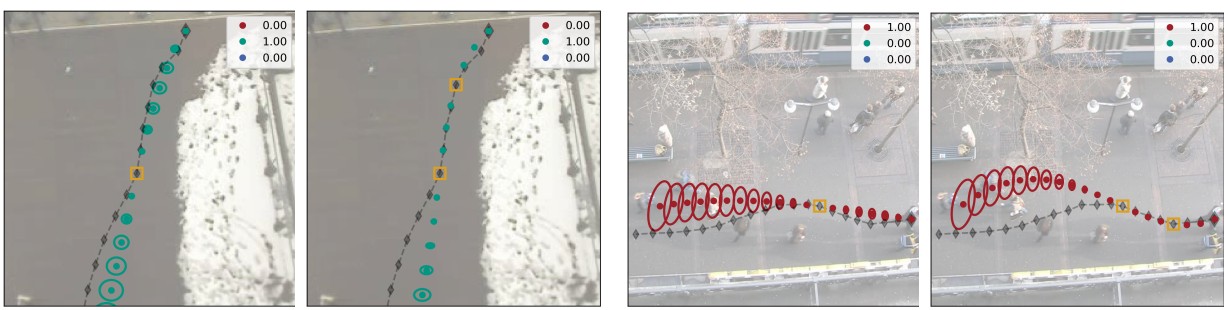

Figure 6: Cases for refinement leading to degraded prediction performance according to the NLL (1 & 2) and ADE (3 & 4). 1 & 3: refinement A, 2 & 4: refinement B. Yellow squares indicate condition points. Ground truth trajectories are depicted in black and start at the top (left example) and on the right site (right example). The first 8 trajectory points are input into the $\mathcal{N}$-MDN.

observation leads to the suppression of wrong modes. Here, the new observation occurs several time steps after the last original input. Using the $\mathcal{N}$-GP, the updated prediction can be directly calculated without requiring an additional pass through the $\mathcal{N}$-MDN.

## 7 Summary

In this paper, we presented an approach for enabling full Bayesian inference without the need for Monte Carlo methods on top of the $\mathcal{N}$-Curve Mixture Density Network ($\mathcal{N}$-MDN), which is a regression-based probabilistic sequence model and outputs (mixtures of) probabilistic Bézier curves ($\mathcal{N}$-Curves). In our approach, the $\mathcal{N}$-MDN is embedded in the GP framework as a generator for prior distributions. For this, we

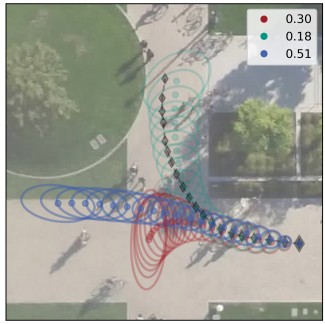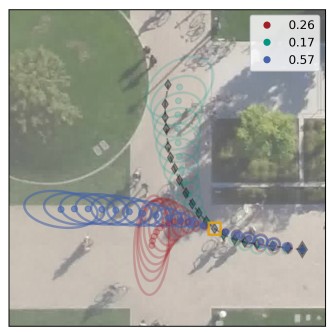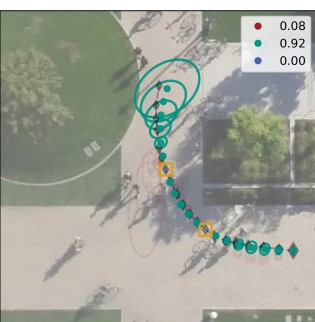

Figure 7: Example for updating the prediction by an $\mathcal{N}$-MDN using the last observed point and an additional point within the predictive time horizon. Yellow squares indicate condition points. Ground truth is depicted in black.

first showed that $\mathcal{N}$-Curves are a special case of Gaussian processes (denoted as $\mathcal{N}$-GP) and then derived mean and kernel functions for the univariate, multivariate and multi-modal cases. In our evaluation on the task of human trajectory prediction, we showed that using the $\mathcal{N}$-GP, predictions generated by an $\mathcal{N}$-MDN can be improved by conditioning on different subsets of the original input. Additionally, we looked briefly into practical applications of the approach, focusing on updating predictions generated by the $\mathcal{N}$-MDN in light of new observations within the predictive time horizon. Using our approach, such updates do not require any additional passes through the $\mathcal{N}$-MDN. Further, missing intermediate observations are inherently handled by the $\mathcal{N}$-GP.

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

## A  Positive Semi-definiteness of the $\mathcal{N} - GP$ Kernel

In this section we provide proof, that the $\mathcal{N}$-GP's kernel function is positive semi-definite, i.e. $k_{\mathcal{P}}(t_i, t_j) \geq 0 \ \forall \ (t_i, t_j)$, with $t_* \in [0, 1]$ and defining Gaussian control points $\mathcal{P} = \{P_0, ..., P_L\}$ with $P_l \sim \mathcal{N}(\mu_l, \sigma_l^2)$. In doing so, we limit the proof to the 1-dimensional case, as it directly translates to the higher-dimensional case as well.

Recalling Eq. 9, the kernel function is defined as

$$k_{\mathcal{P}}(t_i, t_j) = \sum_{l=0}^{L} b_{l,L}(t_i) b_{l,L}(t_j)(\sigma_l^2 + \mu_l^2) + \sum_{l=0}^{L} \left( \sum_{l'=0, l' \neq l}^{L} b_{l,L}(t_i) b_{l',L}(t_j) \mu_l \mu_{l'} \right) - \mu_X \mu_Y,$$

with $\mu_X = \sum_{l=0}^{L} b_{l,L}(t_i) \mu_l$ and $\mu_Y = \sum_{l=0}^{L} b_{l,L}(t_j) \mu_l$. Writing out $\mu_X$ and $\mu_Y$ and combining superfluous summations yields

$$\sum_{l=0}^{L} b_{l,L}(t_i) b_{l,L}(t_j)(\sigma_l^2 + \mu_l^2) + \sum_{l=0}^{L} \left( \sum_{l'=0, l' \neq l}^{L} b_{l,L}(t_i) b_{l',L}(t_j) \mu_l \mu_{l'} \right) - \mu_X \mu_Y \geq 0$$

$$\sum_{l=0}^{L} b_{l,L}(t_i) b_{l,L}(t_j)(\sigma_l^2 + \mu_l^2) + \underbrace{\sum_{l=0}^{L} \left( \sum_{l'=0, l' \neq l}^{L} b_{l,L}(t_i) b_{l',L}(t_j) \mu_l \mu_{l'} \right) - \sum_{l=0}^{L} \left( \sum_{l'=0}^{L} b_{l,L}(t_i) b_{l',L}(t_j) \mu_l \mu_{l'} \right)}_{= -\sum_{l=0}^{L} b_{l,L}(t_i) b_{l,L}(t_j) \mu_l^2} \geq 0$$

$$\sum_{l=0}^{L} \left( b_{l,L}(t_i) b_{l,L}(t_j) \sigma_l^2 + b_{l,L}(t_i) b_{l,L}(t_j) \mu_l^2 - b_{l,L}(t_i) b_{l,L}(t_j) \mu_l^2 \right) \geq 0$$

$$\sum_{l=0}^{L} b_{l,L}(t_i) b_{l,L}(t_j) \sigma_l^2 \geq 0,$$

which is independent of the control point mean values $\mu_l$ and greater or equal to zero for any $t_* \in [0, 1]$, due to $b_{l,L}(t_*) \geq 0$ and $\sigma_l^2 > 0$. Thus, $k_{\mathcal{P}}(t_i, t_j)$ is positive semi-definite for any $t_* \in [0, 1]$ and arbitrary sets of Gaussian control points.

## B  State-of-the-Art Comparison of the $\mathcal{N}$-MDN model

In this section, we provide state-of-the-art comparisons of the underlying $\mathcal{N}$-MDN used in our paper considering several standard benchmarking datasets. Thereby, we aim to achieve a fair and concise comparison between different approaches in the literature, focusing on base sequence models by restricting the model input to raw positional information and disregarding variations of the same base model. Further, we are employing common performance measures and baselines.

### B.1  Human Trajectory Prediction Models

The core component of human trajectory prediction models is given by a base sequence model, which encodes input trajectories, the *observation*, and generates either single trajectories or probabilistic predictions. Taking a range of state-of-the-art deep learning-based prediction models into consideration, these models can be boiled down to few base sequence models. Hence, the most frequently used sequence models are given by *Recurrent Mixture Density Networks* (abbrev.: R-MDN, Graves (2013)), variants of *Generative Adversarial Networks* (abbrev.: GAN, Goodfellow et al. (2014)) and *Variational Autoencoders* (abbrev.: VAE, Kingma & Welling (2014)) combined with a sequence to sequence model Sutskever et al. (2014), as well as *Transformers* Vaswani et al. (2017). Note that due to the existence of many similar models, the references in this list are reduced to the most commonly used foundational models in human trajectory prediction. For a comprehensive overview of existing human trajectory prediction approaches, the reader may be referred to recent surveys, e.g. Rudenko et al. (2020).

## B.2 Performance Measures

In the standard evaluation approach, the designated performance measures are given by the *Average Displacement Error* (abbrev.: *ADE*) and the *Final Displacement Error* (abbrev.: *FDE*), defined as

$$\text{ADE} = \frac{1}{M \cdot N_{\text{pred}}} \sum_{i=1}^{M} \sum_{t=1}^{N_{\text{pred}}} \|\hat{\mathbf{y}}_t^i - \mathbf{y}_t^i\|_2 \tag{22}$$

and

$$\text{FDE} = \frac{1}{M} \sum_{i=1}^{M} \|\hat{\mathbf{y}}_{N_{\text{pred}}}^i - \mathbf{y}_{N_{\text{pred}}}^i\|_2 \tag{23}$$

for a given prediction horizon $N_{\text{pred}}$, a set $\boldsymbol{\mathcal{Y}} = \{\mathcal{Y}_1, ..., \mathcal{Y}_M\}$ of $M$ ground truth trajectories $\mathcal{Y}_i = \{\mathbf{y}_1^i, ..., \mathbf{y}_{N_{\text{pred}}}^i\}$ and corresponding predictions $\hat{\mathcal{Y}}_i = \{\hat{\mathbf{y}}_1^i, ..., \hat{\mathbf{y}}_{N_{\text{pred}}}^i\}$ generated by a given prediction model. The ADE is then defined by the average L2 distance between the ground truth and a corresponding predicted trajectory, while the FDE is defined by the L2 distance between the final ground truth and predicted trajectory points after the prediction horizon. In the case of probabilistic sequence models, which generate a predictive distribution $p(\mathbf{y}_{\{1,...,N_{\text{pred}}\}}|\mathbf{x}_{\{1,...,N_{\text{in}}\}})$, $\hat{\mathcal{Y}}_i$ corresponds to a maximum likelihood prediction given the probabilistic output of the model.

As the ADE and FDE do not provide an adequate measure for assessing the quality of (multi-modal) probabilistic predictions, another performance measure is required for this case. Due to the actual ground truth probability distribution for each time step being unknown, a common choice is given by the *Negative (data) Log-Likelihood* (abbrev.: *NLL*, e.g. Bhattacharyya et al. (2018); Ivanovic & Pavone (2019)), defined as

$$\text{NLL} = \frac{1}{M \cdot N_{\text{pred}}} \sum_{i=1}^{M} \sum_{t=1}^{N_{\text{pred}}} -\log p(\mathbf{y}_t^i|\cdot). \tag{24}$$

Here, $p(\mathbf{y}_t^i|\cdot)$ denotes the predictive distribution for the $t$'th trajectory position as generated by the probabilistic sequence model. Note that the conditional part of this distribution is not given explicitly, as it varies between different models. It is worth mentioning, that sometimes an oracle measure (e.g. Lee et al. (2017)) is used as a sample-based substitute for the NLL. This measure does, however, introduce another hyperparameter, which is why the NLL is preferred in the context of this evaluation.

## B.3 Baselines

In order to provide reference values for comparison, a simple baseline is given for each performance measure. In the case of the ADE and FDE, a simple prediction model is given by a linear extrapolation calculated from a respective observed trajectory. Here, the relative offset $\delta_i = \mathbf{x}_{N_{\text{in}}}^i - \mathbf{x}_{N_{\text{in}}-1}^i$ of the two most recent observations is projected $N_{\text{pred}}$ steps into the future, as these positions are assumed to have the most impact on the future trajectory Schöller et al. (2020). In the case of the NLL measure, a sample-based prediction can be generated for each future position by using a *shotgun* approach Pajouheshgar & Lampert (2018). In this approach, multiple future trajectories are generated by randomly altering the direction and scale of the relative offset $\delta_i$ before projection. The altered offset for each future trajectory is then given by $\mathbf{R}_\alpha \cdot \delta_i \cdot s$ with $\alpha \sim \mathcal{N}(0, \sigma_\alpha)$, $s \sim \mathcal{N}(1, \sigma_s)$ and the matrix $\mathbf{R}_\alpha$ describing a rotation by $\alpha$ degrees. This yields a unimodal probabilistic prediction with a fixed variance for each predicted time step. In the following, $\sigma_\alpha = 15°$ and $\sigma_s = 0.1$ are used.

In addition to these two baselines, a simple LSTM baseline is provided. This mainly has two reasons. On the one hand, the LSTM model is an integral component of multiple sequence models included in the evaluation. On the other hand, it is a widely used baseline next to the linear extrapolation approach.

## B.4 Implementation Details

This section gives a brief overview on implementation details for the sequence models in comparison. The implementations are based on existing approaches, which provide a publicly available implementation. These implementations are adapted to use a common data pipeline. If necessary, components for processing additional cues, such as social context, are removed. The list of approaches the implementations are based on alongside adaptations made is given in Table 2.

| Model | Based on | Adaptations |
|---|---|---|
| R-MDN | ParticleLSTM Hug et al. (2018), Own implementation | - |
| GAN | Social GAN Gupta et al. (2018), Original implementation[10] | Removed social context Data pipeline |
| VAE | LSTM-BMS Bhattacharyya et al. (2018), Original implementation[11] | PyTorch re-implementation Data pipeline |
| Transformer | TF Giuliari et al. (2021), Original implementation[12] | Data pipeline |

Table 2: Sequence model implementations adapted for use in this evaluation.

**R-MDN:** The SMC-based R-MDN variant used in this evaluation is a 1-to-1 sequence model, processing one trajectory point at a time. As such, the model takes a discrete trajectory point as input and outputs the parameters of a Gaussian mixture distribution modeling the next trajectory point. In order to enable the model to generate a multi-modal prediction, multiple points are sampled from the output distribution and fed back into the model. To prevent exponential growth of samples, subsequent output distributions are combined and re-sampled Hug et al. (2018).

**GAN:** For applying a conditional GAN in the context of human trajectory prediction, a sequence processing unit must be incorporated into the model. According to Gupta et al. (2018), a sequence-to-sequence LSTM is built into the generator network and another LSTM encoder is built into the discriminator network. The GAN encodes the observed trajectory and then adds a random noise vector to the encoded representation in order to sequentially generate a prediction. By performing multiple passes through the decoder network using different noise vectors, a sample-based distribution of future trajectories is generated.

**VAE:** Similar to the GAN extension, a sequence-to-sequence LSTM is built into the conditional VAE in order to enable sequence processing. Further, prediction generation works similar to the GAN model by adding a random vector to an encoded representation of an observed trajectory in order to generate multiple future trajectories.

**Transformer:** Although the implementation chosen for this evaluation does not provide a probabilistic prediction model, it is considered in this comparison, as it provides a strong contender to the established LSTM networks built into many human trajectory prediction models. It is an attention-based sequence model, consisting of an encoder, which encodes the entire observed trajectory into a single vector, and a decoder, which sequentially generates one trajectory point at a time, given the encoding.

## B.5 Evaluation Methodology

For achieving a reliable evaluation, a $k$-fold cross-validation is performed on each dataset, in order to cope with unfavorable random training and test splits. In the following, $k = 5$ folds are performed, as it gives a

---

[10]https://github.com/agrimgupta92/sgan
[11]https://github.com/apratimbhattacharyya18/CGM_BestOfMany
[12]https://github.com/FGiuliari/Trajectory-Transformer

good trade-off between error bias and variance Hastie et al. (2009). In compliance with the goal of measuring the raw single dataset performance, all prediction models are re-trained for each fold. As is common practice, prediction models are tasked to predict $N_{\text{pred}} = 12$ steps (4.8 seconds) into the future, given an observation of $N_{\text{in}} = 8$ steps (3.2 seconds).

For generating a maximum likelihood prediction, the output of the probabilistic prediction models in comparison need to be processed in different ways. For the R-MDN, instead of propagating a set of particles, the mean vector of the highest weighted mixture component is fed back into the model in each time step. As the GAN and VAE models generate a set of sample trajectories, the mean position for each time step is used. Finally, for the $\mathcal{N}$-MDN, the mean curve of the $\mathcal{N}$-Curve with the highest mixture weight is used.

Looking at the NLL measure, which requires a probability density function generated by each prediction model, sample-based output is processed by applying a kernel density estimation Scott (2018) using a Gaussian kernel in order to obtain probability density functions for each time step.

## B.6 Results

For the evaluation, multiple output-related configurations are considered for the R-MDN and $\mathcal{N}$-MDN models, controlling the number of mixture components and the $\mathcal{N}$-MDN model's output mode. The configurations are depicted in Table 3.

| Configuration | Description |
| --- | --- |
| R-MDN$_a$ | Outputs a single-component mixture of Gaussians. |
| R-MDN$_b$ | Outputs a 3-component mixture of Gaussians. |
| $\mathcal{N}$-MDN$_a$ | Models observed & future trajectory. Outputs a single $\mathcal{N}$-Curve. |
| $\mathcal{N}$-MDN$_b$ | Models observed & future trajectory. Outputs mixture of 3 $\mathcal{N}$-Curves. |
| $\mathcal{N}$-MDN$_c$ | Models future trajectory and outputs a single $\mathcal{N}$-Curve. |
| $\mathcal{N}$-MDN$_d$ | Models future trajectory and outputs a mixture of 3 $\mathcal{N}$-Curves. |

Table 3: Output-related configurations for the R-MDN and $\mathcal{N}$-MDN models in the evaluation.

Tables 4 – 9 summarize the results of the quantitative evaluation, using a per dataset 5-fold cross validation and the ADE, FDE and NLL performance measures. Accordingly, respective averaged performance values with corresponding standard deviation considering all 5 folds are depicted. It should be noted, that the performance values are not comparable across datasets, due to different image and ground resolutions. In order to make values comparable, datasets would need to be projected into 3-dimensional world space. Additionally, a re-sampling of trajectory points can be necessary in order to match motion profiles.

| Model | ADE | FDE | NLL |
|---|---|---|---|
| Linear | $21.80 \pm 1.60$ | $47.81 \pm 5.41$ | - |
| Shotgun | - | - | $10.07 \pm 2.37$ |
| LSTM | $13.44 \pm 0.97$ | $26.83 \pm 2.52$ | - |
| Transformer | $19.36 \pm 2.16$ | $35.81 \pm 3.67$ | - |
| R-MDN$_a$ | $26.12 \pm 17.42$ | $46.28 \pm 32.50$ | $16.62 \pm 3.67$ |
| R-MDN$_b$ | $16.11 \pm 3.70$ | $28.70 \pm 7.72$ | $2893.14 \pm 5760.97$ |
| VAE | $22.33 \pm 1.82$ | $37.45 \pm 4.91$ | $8.38 \pm 0.16$ |
| GAN | $11.42 \pm 2.18$ | $22.26 \pm 4.48$ | $1084.14 \pm 988.79$ |
| $\mathcal{N}$-MDN$_a$ | $9.51 \pm 0.67$ | $17.41 \pm 1.13$ | $\mathbf{7.25} \pm 0.13$ |
| $\mathcal{N}$-MDN$_b$ | $\mathbf{9.17} \pm 1.28$ | $\mathbf{17.15} \pm 2.89$ | $7.30 \pm 0.30$ |
| $\mathcal{N}$-MDN$_c$ | $10.23 \pm 1.07$ | $18.61 \pm 2.88$ | $7.48 \pm 0.41$ |
| $\mathcal{N}$-MDN$_d$ | $9.87 \pm 1.03$ | $18.28 \pm 3.40$ | $7.27 \pm 0.17$ |

Table 4: Quantitative results of all approaches on the *biwi:eth* dataset for a predictive time horizon of $N_{\text{pred}} = 12$ time steps (4.8 seconds). ADE and FDE errors are reported in pixels. Lower is better for all performance measures.

| Model | ADE | FDE | NLL |
|---|---|---|---|
| Linear | $26.65 \pm 1.24$ | $51.13 \pm 3.29$ | - |
| Shotgun | - | - | $9.87 \pm 1.14$ |
| LSTM | $17.91 \pm 2.16$ | $32.93 \pm 4.50$ | - |
| Transformer | $20.42 \pm 1.48$ | $34.35 \pm 2.95$ | - |
| R-MDN$_a$ | $24.46 \pm 4.92$ | $43.52 \pm 8.28$ | $15.52 \pm 2.93$ |
| R-MDN$_b$ | $19.01 \pm 3.77$ | $33.57 \pm 6.71$ | $12.20 \pm 2.42$ |
| VAE | $20.03 \pm 4.14$ | $35.61 \pm 7.90$ | $8.25 \pm 0.31$ |
| GAN | $15.48 \pm 1.80$ | $26.38 \pm 3.15$ | $20115.70 \pm 38614.18$ |
| $\mathcal{N}$-MDN$_a$ | $16.64 \pm 3.10$ | $30.36 \pm 7.56$ | $8.26 \pm 0.48$ |
| $\mathcal{N}$-MDN$_b$ | $15.46 \pm 2.03$ | $27.28 \pm 3.75$ | $7.96 \pm 0.16$ |
| $\mathcal{N}$-MDN$_c$ | $\mathbf{13.76} \pm 0.94$ | $\mathbf{23.82} \pm 1.90$ | $7.86 \pm 0.18$ |
| $\mathcal{N}$-MDN$_d$ | $15.30 \pm 1.98$ | $26.60 \pm 4.22$ | $\mathbf{7.85} \pm 0.20$ |

Table 5: Quantitative results of all approaches on the *biwi:hotel* dataset for a predictive time horizon of $N_{\text{pred}} = 12$ time steps (4.8 seconds). ADE and FDE errors are reported in pixels. Lower is better for all performance measures.

| Model | ADE | FDE | NLL |
|-------|-----|-----|-----|
| Linear | $21.69 \pm 0.40$ | $46.98 \pm 1.39$ | - |
| Shotgun | - | - | $10.93 \pm 0.89$ |
| LSTM | $23.71 \pm 9.30$ | $49.93 \pm 24.18$ | - |
| Transformer | $27.30 \pm 1.64$ | $50.04 \pm 2.67$ | - |
| R-MDN$_a$ | $20.15 \pm 3.25$ | $38.38 \pm 6.77$ | $14.76 \pm 4.55$ |
| R-MDN$_b$ | $16.86 \pm 0.34$ | $31.18 \pm 0.72$ | $13.31 \pm 2.26$ |
| VAE | $19.21 \pm 0.74$ | $35.49 \pm 1.47$ | $8.24 \pm 0.19$ |
| GAN | $\mathbf{15.59} \pm 0.41$ | $\mathbf{30.11} \pm 0.83$ | $363.05 \pm 456.18$ |
| $\mathcal{N}$-MDN$_a$ | $18.48 \pm 0.51$ | $35.97 \pm 0.74$ | $8.30 \pm 0.03$ |
| $\mathcal{N}$-MDN$_b$ | $19.07 \pm 0.68$ | $36.49 \pm 1.40$ | $8.21 \pm 0.06$ |
| $\mathcal{N}$-MDN$_c$ | $16.60 \pm 0.44$ | $32.12 \pm 0.84$ | $8.04 \pm 0.04$ |
| $\mathcal{N}$-MDN$_d$ | $17.71 \pm 0.43$ | $34.17 \pm 0.84$ | $\mathbf{7.95} \pm 0.05$ |

Table 6: Quantitative results of all approaches on the *crowds:zara01* dataset for a predictive time horizon of $N_{\mathrm{pred}} = 12$ time steps (4.8 seconds). ADE and FDE errors are reported in pixels. Lower is better for all performance measures.

| Model | ADE | FDE | NLL |
|-------|-----|-----|-----|
| Linear | $28.08 \pm 0.33$ | $60.83 \pm 0.73$ | - |
| Shotgun | - | - | $14.97 \pm 1.72$ |
| LSTM | $34.14 \pm 22.66$ | $72.47 \pm 52.74$ | - |
| Transformer | $32.17 \pm 2.94$ | $58.25 \pm 4.76$ | - |
| R-MDN$_a$ | $24.92 \pm 1.98$ | $48.18 \pm 3.59$ | $11.99 \pm 1.37$ |
| R-MDN$_b$ | $21.72 \pm 2.59$ | $41.19 \pm 4.65$ | $11.67 \pm 1.17$ |
| VAE | $23.44 \pm 0.76$ | $43.45 \pm 1.42$ | $9.29 \pm 0.27$ |
| GAN | $\mathbf{20.01} \pm 0.72$ | $\mathbf{39.57} \pm 1.35$ | $26.33 \pm 13.40$ |
| $\mathcal{N}$-MDN$_a$ | $22.34 \pm 0.92$ | $43.20 \pm 2.03$ | $8.54 \pm 0.07$ |
| $\mathcal{N}$-MDN$_b$ | $24.74 \pm 1.30$ | $47.29 \pm 2.90$ | $8.49 \pm 0.06$ |
| $\mathcal{N}$-MDN$_c$ | $20.41 \pm 1.07$ | $40.39 \pm 2.35$ | $8.33 \pm 0.07$ |
| $\mathcal{N}$-MDN$_d$ | $21.41 \pm 1.03$ | $41.55 \pm 2.45$ | $\mathbf{7.98} \pm 0.03$ |

Table 7: Quantitative results of all approaches on the *crowds:zara02* dataset for a predictive time horizon of $N_{\mathrm{pred}} = 12$ time steps (4.8 seconds). ADE and FDE errors are reported in pixels. Lower is better for all performance measures.

| Model | ADE | FDE | NLL |
|---|:---:|:---:|:---:|
| Linear | $28.39 \pm 0.23$ | $60.21 \pm 0.67$ | - |
| Shotgun | - | - | $18.03 \pm 4.00$ |
| LSTM | $27.91 \pm 0.95$ | $56.04 \pm 1.75$ | - |
| Transformer | $52.48 \pm 14.88$ | $97.00 \pm 26.24$ | - |
| R-MDN$_a$ | $50.02 \pm 13.19$ | $93.60 \pm 23.37$ | $14.28 \pm 2.55$ |
| R-MDN$_b$ | $27.58 \pm 4.64$ | $50.11 \pm 8.56$ | $23.47 \pm 23.88$ |
| VAE | $30.75 \pm 1.04$ | $55.93 \pm 1.83$ | $8.88 \pm 0.09$ |
| GAN | $\mathbf{19.10} \pm 0.58$ | $35.45 \pm 0.96$ | $33.18 \pm 4.71$ |
| $\mathcal{N}$-MDN$_a$ | $25.24 \pm 1.24$ | $48.28 \pm 2.80$ | $9.13 \pm 0.05$ |
| $\mathcal{N}$-MDN$_b$ | $25.33 \pm 2.34$ | $48.69 \pm 5.28$ | $9.06 \pm 0.09$ |
| $\mathcal{N}$-MDN$_c$ | $22.31 \pm 1.15$ | $41.60 \pm 1.58$ | $8.89 \pm 0.07$ |
| $\mathcal{N}$-MDN$_d$ | $19.43 \pm 0.55$ | $\mathbf{35.37} \pm 1.10$ | $\mathbf{8.46} \pm 0.04$ |

Table 8: Quantitative results of all approaches on the *sdd:bookstore03* dataset for a predictive time horizon of $N_{\mathrm{pred}} = 12$ time steps (4.8 seconds). ADE and FDE errors are reported in pixels. Lower is better for all performance measures.

| Model | ADE | FDE | NLL |
|---|:---:|:---:|:---:|
| Linear | $36.25 \pm 0.79$ | $75.93 \pm 2.09$ | - |
| Shotgun | - | - | $16.33 \pm 1.30$ |
| LSTM | $40.52 \pm 9.44$ | $85.83 \pm 23.63$ | - |
| Transformer | $119.70 \pm 0.68$ | $222.18 \pm 1.39$ | - |
| R-MDN$_a$ | $90.86 \pm 23.89$ | $168.85 \pm 43.22$ | $20.43 \pm 4.03$ |
| R-MDN$_b$ | $44.18 \pm 2.32$ | $84.29 \pm 4.17$ | $13.46 \pm 0.74$ |
| VAE | $41.39 \pm 2.53$ | $82.21 \pm 5.21$ | $9.38 \pm 0.09$ |
| GAN | $\mathbf{28.84} \pm 1.11$ | $\mathbf{57.56} \pm 2.92$ | $20.48 \pm 5.26$ |
| $\mathcal{N}$-MDN$_a$ | $35.83 \pm 1.03$ | $72.74 \pm 2.49$ | $9.82 \pm 0.05$ |
| $\mathcal{N}$-MDN$_b$ | $37.62 \pm 2.57$ | $76.31 \pm 6.07$ | $9.80 \pm 0.07$ |
| $\mathcal{N}$-MDN$_c$ | $34.30 \pm 1.11$ | $69.87 \pm 2.37$ | $9.63 \pm 0.02$ |
| $\mathcal{N}$-MDN$_d$ | $29.68 \pm 1.18$ | $58.69 \pm 3.13$ | $\mathbf{9.11} \pm 0.01$ |

Table 9: Quantitative results of all approaches on the *sdd:hyang00* dataset for a predictive time horizon of $N_{\mathrm{pred}} = 12$ time steps (4.8 seconds). ADE and FDE errors are reported in pixels. Lower is better for all performance measures.

