# OpenReview forum: "Bayesian Inference for Sequence Mixture Density Networks using Bézier Curve Gaussian Processes"
_TMLR — Rejected by TMLR_

### Review · Reviewer_5PUr · 2023-09-15

**Summary Of Contributions:**

The paper plays around with the potential similarity between probabilistic Bezier curves and GPs. Using Hug et al. 2020 as the main reference, the model based on mixture density networks (MDNs) and Bezier curves is first linked as a special variant of GPs. Later, a multivariate and multi-modality extension is presented to later follow on the Bayesian inference.

**Audience:**

Yes

**Claims And Evidence:**

No

**Requested Changes:**

I would like to see changes on the three points raised before (**P1**,**P2** and **P3**). However, I advice the authors that this would require major changes in the paper as it can be understood from my concerns on the novelty, the experiments and the theoretical connection with GPs.

**Strengths And Weaknesses:**

The paper is clear and decently written -- in that sense is easy and enjoyable to read. However, there are strong weaknesses that I cannot omit. In my opinion, the current manuscript has three main problems:

**P1)** The similarity with Hug et al. 2020 is extremely high, as there is not much novelty compared with what is presented in there. Importantly, in that paper, it is clearly described that certain trajectories might be considered as realisations of a stochastic process and in section 3 -- it explicitly says that such stochastic processes are modelled using Bezier curves. In that way, the presentation of the model, parameterisation, assumptions and estimation are almost identical. Even in the section 3, it is considered the case of approximating multi-modal processes and mixtures of probability distributions, which is supposed to be the main contribution of the current submission.

**P2)** From my perspective, I first felt that the affine transformation in section 3 is the primarily argument used to link Bezier curves and GPs in this case. However, I think this is not sufficient, and the reader feels that this connection is not very clear from the text as sometimes is described as a variant of GPs, others as a non-stationary version or just a GP with an 'implicit' covariance function. In my opinion, and also having the example of Jorgensen & Osborne in mind, this connection should not be just done in this simple way. So I encourage the authors to be more clear and concise on this connection, link to similar works and overall to provide convincing arguments, which I think are currently lacking.

**P3)** Perhaps, the most important problem for the novelty, is that Jorgensen & Osborne recently covered the connection between Bezier curves and GPs in a thorough way. I am not saying that this avoids any progress of the current ideas, but the paragraph before section 6 arguing that the differences are due to the consideration of spatial inputs and variational inference is just not enough to me. Mainly, because this point is only cited at the end and no comparison is included in the experiments, which I find not problematic..

----------------------
> *Hug et al. 2020 / Introducing Probabilistic Bezier Curves for N-Step Sequence Prediction / AAAI 2020*

> *Jorgensen & Osborne / Bézier Gaussian Processes for Tall and Wide Data / NeurIPS 2022*

---

> ### Author Response · Authors · 2023-09-26
> **Response to Reviewer 5PUr (Part 1)**
>
> First of all, we want to thank the reviewer for taking their time reviewing our manuscript. In the following, we want to address the raised concerns individually. In order to make sure there is no misunderstanding on our side, we will start each paragraph by paraphrasing the respective concern.
>
> **P1) Our manuscript is too similar to Hug et al. (2020) and there is not much novelty compared to it:** As the reviewer has pointed out, there are some natural similarities between our manuscript and Hug et al. given we are incorporating their N-MDN neural network in our approach and extend their concept of N-Curves (probabilistic Bézier curves). Following that, it is important to highlight, that the N-Curve concept and the N-MDN model are two separate contributions: N-Curves as a representation of a probabilistic sequence model and the N-MDN neural networks as a means of learning and applying those curves to the task of trajectory prediction. Now, in our manuscript, we take the concept of N-Curves and extend the theory surrounding them by establishing a link to Gaussian processes, providing a proof of equivalence, and deriving the GP mean and kernel functions for a Gaussian process (N-GP) induced by a given N-Curve for the univariate, multivariate and multi-modal cases. We thus show that an N-Curve is actually a Gaussian process and in that an alternate representation of which, which does not require explicit mean and kernel functions. We further extend their trajectory prediction model by re-purposing their N-MDN neural network (which is a probabilistic sequence prediction model) as a data-dependent generator for N-GP prior distributions. This line of research was not included in Hug et al. (2020) nor indicated. We acknowledge though that there can be some confusion caused by the basic assumption of probabilistic sequence models generally being models for some unknown stochastic process, which generates a given training dataset. This is, however, to be distinguished from the Gaussian process induced by an N-Curve, which is simply a variant or type of Gaussian process (which could in turn be used to approximate an unknown data-generating stochastic process). *We will state our contributions that add to the work of Hug et al. (2020) in our related works section, in order to clarify the novel contributions in our manuscript.*
>
> **P2) The connection between N-Curves and Gaussian processes via an affine transformation is too simple and terminology surrounding the N-GP is not clear at all times:** Throughout the paper we are using different attributes to describe the N-GP and we acknowledge that this could be confusing to the reader. *We will attempt to clear some of the uncertainties in the following and will revise the manuscript accordingly.* Starting with the affine transformation mentioned by reviewer, this is part of our equivalence proof we have provided to show that N-Curves are actually a specific representation of a specific group of Gaussian processes, i.e. non-stationary Gaussian processes. Here, the N-Curve representation does not require an explicit kernel (or covariance) function, as the correlation between points on the curve emerges from the geometric constraints of the underlying Bézier curve. We thus say that the kernel function is given implicitly. Circling back to the affine transformation, when proving an N-Curve is actually a Gaussian process, we define a set of points on the N-Curve as the set of elements of a Gaussian process, which have to fulfill the GP property (see e.g. Rasmussen & Williams (2006) or MacKay (2003)) in order to actually define a Gaussian process. One way to obtain the set of points is given by using the affine transformation, which directly yields the GP property to hold true, as affine transformations of Gaussian variables (the control points in this case) are again Gaussian, and thus prove the equivalence. With this link established, we can then convert the N-Curve representation into the common GP representation using a mean and kernel function, we then proceed to derive mean and kernel functions for different use-cases as the remainder of section 3.

---

> ### Author Response · Authors · 2023-09-26
> **Response to Reviewer 5PUr (Part 2)**
>
> **P3a) The novelty over Jorgensen & Osborne (2022) is not quite clear:** While there are some obvious similar aspects about their work and our manuscript considering both approaches share the same core idea of employing a Gaussian process induced by a probabilistic Bézier curve, we would argue that their focus and general approach is different enough to consider both works as complementary, as the number of differences outweighs the similarities. Apart from their focus on spatial input, one of their main concerns is given by scalability, which is a frequent issue with Gaussian processes. In our manuscript, however, we rather focus on the derivations of mean kernel functions for different use-cases. Thereby, we also explore some properties of the N-GP and show surface-level comparisons to other Gaussian process kernels. We further provide an equivalence proof for N-Curves being Gaussian processes, which is indicated, but not provided in their paper. Lastly, most models in probabilistic sequence modeling employ a variational approach, like Jorgensen & Osborne also do. We, on the other hand, provide a regression-based learning approach by incorporating the N-MDN neural network from Hug et al. (2020) into the N-GP learning approach, thus providing an alternative learning strategy. *We will revise our manuscript and extend the respective paragraph in section 6 in order to make the differences between their paper and our manuscript more clear.*
>
> **P3b) The approach by Jorgensen & Osborne (2022) is not considered in our evaluation:** The primary purpose of our evaluation section is to examine the viability of the re-purposed N-MDN, embedded into the GP framework in order to a) improve its prediction performance and b) enable additional Bayesian inference tasks, which were not possible in the original model. A comparison to Jorgensen & Osborne (2022) does not fit that purpose well and would rather disperse the focus of the evaluation section. Further, with their model being purposed for spatial rather than sequential input, the reliability of their model for sequence modeling is not quite clear.
>
> **References:**
>
> Hug et al. 2020 / Introducing Probabilistic Bezier Curves for N-Step Sequence Prediction / AAAI 2020
>
> Jorgensen & Osborne / Bézier Gaussian Processes for Tall and Wide Data / NeurIPS 2022
>
> Rasmussen & Williams / Gaussian processes for machine learning / MIT press 2006
>
> MacKay / Information theory, inference and learning algorithms / Cambridge university press 2003

---

### Review · Reviewer_3T5X · 2023-10-14

**Summary Of Contributions:**

The paper develops the use of probabilistic Bezier curves (N-curves) for modeling sequences. In this work, the authors show that N-Curves are special cases of Gaussian Processes (GPs), and derive the mean and covariance functions for the univariate, multivariate and multi-modal cases.

The paper then proposes a probabilistic model that combines a mixture density neural network (MDN) with N-curves, which is a more complex model than a single N-curve alone.

Experiments demonstrate the ability of the model to predict multiple modes in trajectory prediction tasks.

**Audience:**

Yes

**Claims And Evidence:**

No

**Requested Changes:**

Requested changes:
- In the introduction, the machine learning problem is unclear. The authors discuss VAEs and GANs, but it's not clear if they are discussing autoregressive models of sequences, where a conditional VAE (or GAN) is used to predict the next sequence element (or multiple sequence elements like in the N-MDN), or if the entire sequence is being generated at once.
- Introduction p2: "due to MDN-based models being restricted to learning max likelihood estimates of the target distribution, such models are potentially less expressive as a probabilistic model, e.g. in terms of representing model uncertainty." This doesn't make sense. The MDN model is less expressive because it is a simpler model, with the output distribution constrained to be a mixture of Gaussians.
- Introduction p4: "in this paper we narrow the gap between MDNs and the approximate Bayesian models in terms of expressiveness by establishing a connection between N-MDNs and the Gaussian Process framework." I disagree with this statement. Showing that N-MDN is a special case of a mixture of GPs tells us that the model is less flexible than a more general mixture of GPs. It certainly helps us understand the flexibility of the model, but this does not narrow any gap.
- Introduction contribution 3: "[the new model] combines the stability and low computational complexity of N-curve based MDNs with the expressivity and flexibility of a Gaussian process" This doesn't make sense, since a N-curve is shown to be a Gaussian process, shouldn't the new model be more flexible than a GP alone?
- Equation 1: Typo where normal distribution is missing an N.
- Section 2 last paragraph: "which avoids the need for Monte Carlo methods in multi-modal inference by directly parameterizing..." I think it would be more clear to say "which avoids the need for Monte Carlo methods in sequence generation by directly".
- Section 2: The N-MDN model is referenced throughout the paper but is never defined carefully. I think it should be defined in Section 2 with the other models.
- Section 3: It is shown that an N-Curve is a special case of a GP, and the authors introduce the term N-GP to refer to this, but then they switch between these three terms without any particular reason. I found this confusing.
- Section 4: The description here is very confusing, and I'm not sure what the authors are proposing.

**Strengths And Weaknesses:**

Strengths:
- The idea of using N-curves to model sequences is interesting, and showing how they are a special case of Gaussian Processes (for multiple cases) is a useful contribution.
- A key feature of the model is the ability to have multiple modes in the output probability distribution, making the model more expressive than a simple GP.  However, the number of modes is still limited, making it less expressive than generative neural network models like VAEs or GANs (and thus less likely to overfit). It is useful to have this type of model option.

Weaknesses:
- Many of the descriptions of the models are imprecise and unclear. It would be hard for people to implement this model.
- I disagree with some of the motivations given for developing this model. Why is this better
- The closest thing would be a mixture model of GPs, but the authors don't say much about this comparison.

---

> ### Author Response · Authors · 2023-10-23
> **Response to Reviewer 3T5X (Part 1)**
>
> First of all, we want to thank the reviewer for taking their time reviewing our manuscript. In the following, we want to address the raised concerns individually. In order to make sure there is no misunderstanding on our side, we will start each paragraph by paraphrasing the respective concern. Further, we will address the stated requests changes and the weaknesses separately, starting with the list of requested changes.
>
>
> ## 1. Requested Changes
>
> **The machine learning problem in the introduction is unclear and thus it is not clear if autoregressive variants of VAE/GAN (conditional) for sequences are discussed:** As pointed out correctly, we are actually referring to conditional VAE/GAN variants, which incorporate a seq2seq structure for encoding and generating sequences in an autoregressive manner. *We will make this clearer in the introduction and add appropriate references.* In order to elaborate a little more on that, we only consider the autoregressive variants, as these are the common choice in the context of trajectory prediction models, which is also true for the recurrent MDN variant and why one of the strong points of the N-MDN is getting rid of the autoregressive structure. This may seem like an inferiority of the N-MDN in theory with respect to its inability of generating sequences of any length, but in practice cVAE and cGAN implementations are also bounded by the sequence length chosen in training, as the model is commonly evaluated on predicted sequences of length N rather than on a per element basis. Looking at the recurrent MDN, which is usually evaluated on a per element basis, it is oftentimes prone to heavy error propagation on the occurrence of outliers when propagating discrete multi-modal distributions over multiple time steps. Regarding the machine learning problem, while in the current version of the manuscript we are writing about probabilistic sequence modeling in general, we are mostly referring to probabilistic sequence *prediction* throughout the paper. Thus, we agree that it will help to address the sequence prediction problem directly in the introduction and *we will add it to the paper.*
>
> **The MDN model is less expressive because it is a simpler model with its output distribution constrained to be a mixture of Gaussians and the argument that this is due to the model learning maximum likelihood estimates of the target distribution does not make sense:** It is correct, that the MDN model is less expressive than the other neural models mentioned in our introduction (namely BNN, VAE and GAN) because of its simplicity and restriction to GMM output, and we acknowledge this to be a more obvious argument and *will replace it in the text.* We would still argue that our statement is valid and want to explain it briefly: The example for inferior expressiveness that we have chosen – the capability of representing model uncertainty – is in reference to BNNs, which are well capable of this, while MDNs aren’t, because these are deterministic networks, which learn point estimates for their output in a maximum likelihood fashion. Following this, admittedly the real issue with our original statement is, that even with our extension to the N-MDN, the new model still is not capable of representing model uncertainty, making this an unfortunate example that is better off replaced.
>
> **Connecting MDNs with GPs does not narrow any gap:** While we acknowledge, that there is something not quite intuitive about our statement and we will change it in our manuscript, we would argue that it is not wrong either. What we really intended to state is that (N-)MDNs will become more expressive and flexible by employing our proposed extension. As these models (MDNs) are less expressive than e.g. BNNs, we pictured this “gap” in expressiveness to become narrower, as the “level of expressiveness” associated with the MDN increases by our approach. The real issue with this is, that neither the “level of expressiveness” nor the difference in that for two models can really be quantified, making it rather confusing and not too well suited for building a chain of reasoning. Thus, looking at it from this perspective, we would argue that the statement is not wrong, but unintuitive and should be replaced with something more clearly reflective of what our proposition achieves. *We thus propose to change this statement to clearly state that our paper enhances N-MDNs in different ways.* As a sidenote, we want to clarify, that it is not the N-MDN that is a special case of a mixture of GPs, a mixture of N-Curves is. An N-MDN is just a way of estimating the parameters of an N-Curve mixture from data. We will make this more clear in the updated version of the paper.

---

> > ### Author Response · Authors · 2023-10-23
> > **Response to Reviewer 3T5X (Part 2)**
> >
> > **We have stated, that the combined model combines the stability of low computation complexity of […] MDNs with the […] flexibility of a GP. Why is the combined model not more flexible than a GP alone, despite an N-Curve is shown to be a GP?** We acknowledge that we will have to *revise the paper and make sure, that the distinction between N-Curves, the N-MDN model and our proposed combined model is more clear.* The “starting point” of our paper is the (N-)MDN. This is a neural network-based prediction model, taking an observed sequence as input and generating a prediction of target sequences in terms of an N-Curve (mixture), which is stable during training and in general low-cost in computational complexity. From there, we are adding the capabilities of the GP framework to it, by transitioning its output to be a Gaussian process (prior), or a mixture thereof. This transition happens in a post-hoc fashion. We thus rather gain the level of flexibility a GP has, but don’t surpass it.
> >
> > **Equation 1 contains a normal distribution missing an N:** *Thanks for pointing that out, we will add it in.*
> >
> > **It would be clearer to say “[…] avoids the need for MC methods in sequence generation […]” rather than “[…] avoids the need for MC methods in multi-modal inference […]”:** Thanks for the suggestion, we think it would be best to combine both versions in order to keep some details about this statement with respect to the recurrent MDN, where it is not sequence generation in general that requires MC methods, but only the generation of multi-modal predictions, where a discrete distribution of sequence elements is propagated in an autoregressive manner. For simple unimodal sequence generation, it is sufficient to propagate the mean vector (and covariance matrix) of each (timely) output, in which case the need for MC methods vanishes.
> >
> > **The N-MDN model is referenced throughout the paper but is never defined carefully and it should be defined in section 2:** We have actually introduced it very briefly at the end of section 2. We admit though that it doesn’t stick out enough considering its importance and *we will expand the introduction of the model.*
> >
> > **The text switches between the terms N-Curve, GP and N-GP without any particular reason, which is confusing:** Thanks for pointing that out, *we will revise the paper and look for unnecessary switch-ups in terminology and make the distinction between the introduced concepts more clear*, as these terms are in general not intended to be interchangeable. By showing that an N-Curve is a special case of a GP (or rather an alternative representation without explicit mean and kernel functions), the term “N-Curve” becomes ambiguous, by referring to a certain type of GP and a polynomial curve with variance in its control and curve points at the same time. While the latter is the original definition of the term given in Hug et al. (2020), we have chosen to introduce the term “N-GP” for referring to the definition as a GP. This was intended to enable us to refer to either concept without ambiguity. Further, sometimes we are referring to GPs in general, which is where we are just using the term “GP”.
> >
> > **The description in section 4 is confusing and the propositions are hard to follow:** In essence, section 4 describes how we intend to enhance MDNs for use in probabilistic sequence modeling, by utilizing the N-MDN as a stepping stone into the realm of GPs and thus to enable GP-based Bayesian inference on top of an MDN-based sequence model in a post-hoc manner. This is achieved by employing an N-MDN, which outputs an N-Curve (mixture), as a data-dependent generator for GP prior distributions. In detail, this translates to a) the N-MDN is learned from data, i.e. adapted to a specific dataset, b) from an output N-Curve (mixture), the mean and kernel functions of the corresponding Gaussian process are calculated, these yield a prior distribution and c) using this prior, different posterior (predictive) distributions can then be calculated. In addition to that, section 4 elaborates on possible real-world use cases of this combined approach in the context of an exemplary sequence prediction task (our proof of concept). Finally, for completeness we have decided to also include some technical details, like the loss function for training the N-MDN (and only the N-MDN alone) and how the use cases are translated to the combined model, providing mathematical background for each use case and a schematic of how the N-MDN and GP components work together. This ultimately leads to a section packed with condensed information and *we will revise the section in an effort to make it easier to follow and more intuitive.*

---

> > > ### Author Response · Authors · 2023-10-23
> > > **Response to Reviewer 3T5X (Part 3)**
> > >
> > > ## 2. Weaknesses
> > >
> > > **Many descriptions of the models are imprecise and/or unclear:** Also referring to our answers in the previous section, we expect our manuscript to improve on this by employing the following updates: 1) Extending the description of the N-MDN, 2) introducing the terms N-Curve, N-MDN and N-GP more carefully, and 3) revising section 4 to be easier to follow. We expect revisions of our phrasing to be sufficient for the most part in achieving these updates (we don’t expect the content itself to require major changes, making it viable for revision).
> > >
> > > **Why is our model better:** In order to answer this question, we believe that we have to state the point of reference more clearly and will check how we can put more emphasize on the fact that we are trying to enhance (N-)MDNs for use in probabilistic sequence prediction (or modeling in general) by enabling GP-based inference on top of an N-MDNs output. Given that, our evaluation section is set to answer the question why our combined model is better than the N-MDN model alone, looking at different use cases discussed in section 4. Here, our evaluation has shown, that the proposed combined model provides better prediction performance while also being more flexible in terms of prediction updates. We will revise the evaluation’s introductory section in order to state its purpose more clearly. Further we will check the concluding summary section to better summarize the benefits of our model.
> > >
> > > **We don’t say much about the comparison with mixture of Gaussian process (experts) model:** While we agree, that in-depth comparisons with other GP/kernel models (including the mixture of GP experts model) are an important topic in the general context of N-GPs, we believe that it is out of scope of the presented paper, which is also why we have decided to not include a detailed comparison in this direction and only provide a minor qualitative comparison to 2 standard kernels (linear and rbf). Thus, in the following we will explain our justification for this decision. First, with proposing an alternate formulation for Gaussian processes based on probabilistic Bézier curves (N-Curves), the main focus of this paper is to (a) present the derived mean and kernel functions in order to lay a foundation for future research focused in different directions, (b) establish a connection between N-GPs and regression-based probabilistic neural models, i.e. the (N-)MDN, in order to provide an effective way for parameter estimation and at the same time enrich the MDNs probabilistic expressiveness, and (c) supply a proof of concept building on a real-world task to showcase the suitability and viability of the combined approach. Considering this, we believe that branching out into a comprehensive section about the connection between N-GPs and previous GP and kernel approaches would disperse the focus and general flow of the paper. Further, in order to keep the paper at a reasonable length, we would have to resort to a rather shallow section on this topic, which would in the worst case not add much value to the paper besides an extended list of references. Putting it the other way around, we think that this comparison is in need of detailed comparisons including the mathematical properties of different models, their pros and cons, etc., which cannot be provided in a shallow comparison. Second, leading up to the proof of concept, the content of the paper is, while still trying to be generally applicable, also biased towards the trajectory prediction task, where GP approaches are overshadowed by deep learning-based probabilistic models as indicated in our introductory section. With this in mind, putting too much emphasize on previous GP-based approaches does not fit well into the overall context of the paper. Nonetheless, we want to emphasize the importance of this topic in the general context of N-GPs, that should be examined in detail in follow-up work.

---

### Review · Reviewer_dbPz · 2023-10-16

**Summary Of Contributions:**

In this paper, the authors propose three different results:

1. Proving that \matcal{N}-curve is a Gaussian process.
2. Its extension to multimodal output distribution.
3. Its application to trajectory prediction.

The results in 1 and 2 are clear. Even when the math is cumbersome and faulty. How the authors estimate the trajectory in the experimental section is unreproducible. It is unknown what the input and outputs to the model are and which parameters are being fit.

**Audience:**

No

**Broader Impact Concerns:**

no need

**Claims And Evidence:**

No

**Requested Changes:**

I do not expect to see any changes in this paper, as I think there is nothing that can be done to make it a publishable result.

**Strengths And Weaknesses:**

---

I do not see any strength in this paper, and the material in many places is non-scientific. The paper has many little mistakes from the start. For example, in the abstract, the authors say that the predominant approach to processing time order information is recurrent neural network, without mentioning Kalman filtering or HMMs is at least surprising. Also, in the abstract, the authors say that GANs are a Bayesian approach, showing they do not know the difference between a generative and a Bayesian model.

The proposed \matcal{N}-curve in Section 2.2 is obviously a kernel because the samples are Gaussian distributed and the covariance function is positive semidefinite if mu_l =0. For nonzero mu_l it will depend on the mu_l values, but I believe the authors set all their mu_l to 0. The authors do not prove that (9) is a positive semidefinite kernel. They do not even know they have to prove it. Given that b_{l, L} are polynomials k_p(t_i,t_j) is a linear combination of polynomials kernels for mu_l = 0, the kernel is positive semidefinite. Moreover, k_p(t_i,t_j) is a polynomial kernel. This result is self-evident from Figures 1 and 2, where the first kernel uses L=2, and leads to a linear kernel. The second kernel is a 9th-degree polynomial.

In Section 4.2, the authors proposed equation 19 to estimate the parameters, but we do not know which parameters are being estimated and what kind of initialization is used to estimate those parameters.  We need to estimate one sigma (and one mean) per data point and Gaussian mixture. We also need to estimate the mixture components. If we have N =20 data points and three mixture components, we need to estimate 60 sigma values, 60 mu values, and 3 mixture components.

When I look at Figures 5 and 6, I do not know what data the model is being conditioned on. For example, in Figure 5 (top left) is the only known data at the bottom point? In that case, how does the algorithm know the direction of the sequence? If it knows more points from the sequence why the confidence interval is so bad? Similar questions can be asked about the rest of the results.

In the appendix other methods are used for comparison, but I have no clue how GANs, VAEs, or transformers are used. It is impossible to evaluate the tables the authors are showing.

Finally, Figures 3 and 4 depict an LSTM that is not explained in the paper.

---

> ### Author Response · Authors · 2023-10-18
> **Response to Reviewer dbPz (Part 1)**
>
> We thank the reviewer for their feedback. In the following, we want to address their comments one paragraph at a time. We will start each paragraph by paraphrasing the respective comment and include a suggestion on how we intend to improve the paper accordingly.
>
> **Kalman filters and HMMs should be listed alongside models building on RNNs as predominant probabilistic sequence models:** Without meaning to take away from the importance of Kalman filters and HMMs, we want to explain our reasoning regarding our statement. While we obviously cannot know about every application domain, we think it is fair to say, that in many probabilistic sequence modeling tasks, especially those concerned with probabilistic multi-step sequence prediction – which this paper is biased towards due to our chosen proof of concept – models based on Recurrent Neural Networks (especially LSTM-based networks) are way more common than non-neural models for several years now (there are even multiple papers presenting neural network-based versions of the Kalman filter for filtering problems). Further, Kalman filters are a great tool for filtering problems, but not as universally applicable like for example RNNs. A relevant contender to the neural network-based approaches is given by GP-based approaches, which we also mention in our introduction. This, however, does not take away from the prevalence of neural network-based approaches in all sorts of application domains.
>
> **GANs are a Bayesian approach:** First of all, we agree that GANs are not a Bayesian approach. We have, at some point, summarized VAEs and GANs under the umbrella term *approximate* Bayesian approaches, which we acknowledge to be an unfortunate term and potentially misleading. The term was motivated by the similarity of VAE and GAN models providing the possibility of sampling from an unknown data distribution p(x) without knowing about it explicitly (where the GAN is in essence an implicit density model and the VAE a latent variable model). Architecturally, both models are very similar in their generative parts, employing a transformative approach, where samples of a simple distribution (commonly unit Gaussian) are transformed into a sample-based representation of a more complex probability distribution. The training setup is vastly different though. Now, the term of “approximate Bayesian approach” originated from the interpretation of p(x) as part of Bayes theorem and the utilization of a sampling-based approach to infer p(x), thus we termed it approximate Bayesian, as in approximate inference (where VAE uses variational inference for example). *In order to correct this misleading terminology, we propose to update the abstract and the introduction to refer to VAE and GAN as “transformative probabilistic models”.*
>
> **In the kernel function, all $\mu_l$ are set to 0 and a proof for positive semi-definiteness is missing:** Starting off with the assumption that all $\mu_l$ are set to 0, this is not the case and also wouldn’t make sense in our intended use case for the N-GP, where the mean and kernel functions are equally important. The reason for this is, when modeling sequences of n-dimensional vectors (e.g. a path in 2d space), the mean function basically defines a Bézier curve the sequence of mean vectors moves along through space. Now, when setting all $\mu_l$, which are the mean vectors of the control points, to 0, the curve would collapse into a single point and we weren’t able to model a sequence of spatial positions anymore. With regards to the positive semi-definiteness of the kernel, the inequality $k(t\_i,t\_j) \geq 0$ reduces to $\sum^{L}\_{l=0} b\_{l,L}(t_i) b\_{l,L}(t_j) \sigma^2\_l \geq 0$, which is always non-negative and not dependent on the actual values of $\mu_l$. *However, we acknowledge that a positive semi-definiteness proof is important to provide in this case, we thus add this proof to the appendix section of the paper and add a reference to the main body of the paper.*

---

> ### Author Response · Authors · 2023-10-18
> **Response to Reviewer dbPz (Part 2)**
>
> **It is not clear which parameters are estimated in equation 19 (sec 4.2):** In section 4.2 we provide, for completeness, the training approach for the N-MDN model with the respective loss function. As briefly summarized in section 2.3, the N-MDN model is comprised of an LSTM network for encoding a given input sequence into a vector v, which is then processed by the feed-forward MDN, which outputs a mixture of N-Curves, representing a distribution over the input sequence and a fixed number of subsequent future sequence elements. The parameters to be estimated are then the parameters of the neural network and the loss function is defined in terms of the data log-likelihood over the n-curve mixture distribution, given a set of sequences (trajectories in this case). For training, each trajectory is split into an observed portion – the first N_in steps – and a to be predicted portion – the remaining N_pred steps – where the encoding LSTM receives the observed portion as input and the output N-Curve mixture covers all N_in + N_pred steps. *We will extend our summary of the N-MDN model in section 2.3, in order to make its architecture and output more clear.*
>
> **It is unknown what the input and outputs of the model are and thus how trajectories in the experimental section are estimated is hard to follow:** We think that this can also be attributed to our summary of the N-MDN model being too brief and *we will extend it accordingly.*
>
> **Figures 5 and 6 are not clearly explained:** The first and longer part of our evaluation focuses on the refinement use case defined in section 4.1, where, given an input trajectory, a prediction for future trajectories in terms of an N-Curve mixture (as output by an N-MDN) is refined by first calculating an N-GP (mixture) from the N-Curve mixture and then conditioning this N-GP on different observed trajectory points. Figures 5 and 6 show how this refinement procedure impacts a probabilistic prediction generated by an N-MDN. A schematic for the refinement process is given in section 4.3. *We will revise section 4 to make the use cases for the evaluation more clear.*
>
> **Figures 3 and 4 depict an LSTM that is not explained in the paper:** The LSTM depicted in Figures 3 and 4 is the LSTM sequence encoder incorporated in the N-MDN model, which is in charge of encoding a given sequence into a single vector, such that it can be further processed by the MDN, which follows the LSTM in the model’s computational path. *We acknowledge that this might not be clear enough from our brief summary of the N-MDN given in section 2.3 and we will expand the corresponding paragraph in order to make this more clear.*
>
> **Appendix A is missing a description of how VAE, GAN and Transformer models are working in a trajectory prediction context:** It is correct, that a description of each model as a trajectory prediction model is missing in the appendix and *we propose to add a brief description for each respective model.* To directly give a brief summary in this answer, the GAN and VAE models include an LSTM encoder (like the N-MDN), which encodes a given trajectory, and another LSTM as a decoder, which decodes the encoded observation with noise added to it (according to the respective Generative model networks) into a predicted trajectory of length N_pred, trying to predict the subsequent N_pred steps from the given input. Repeating the decoding process with different noise values for a fixed number of times yields a discrete distribution over the future trajectories. In the case of the Transformer, it takes a given trajectory as input and predicts a single future trajectory (usually the most probable extension of the given trajectory). Although the Transformer is not a probabilistic model in this case, it is included because of the raise in interest in the Transformer model in human trajectory prediction.
>
> **The math is cumbersome and faulty:** This is not really an answer, but a question to the reviewer. Could you please point us to where the math is faulty, such that we can make an effort to correct it?
>
> *As a final remark, we would like to address the review moving subliminally on an unprofessional and personal level, which we do not appreciate. The points of criticism mostly do not touch the core of the work and are in many parts only poorly substantiated, based on simple formalisms resolvable by minor changes and in disregard of information given in the text – we acknowledge though, that some descriptions can be improved for clarity. Nonetheless we are open to engage in constructive discussion with the reviewer based on the reviewers comments and our response.*

---

### Author Response · Authors · 2023-10-24
**Updated version of our manuscript is now available**

Thanks again for your feedback, we have just uploaded an updated version of our manuscript.

In an effort of improving the overall clarity of the paper and making it more self-contained, we have conducted slight adjustments to the following sections (primarily): Abstract, section 1 (several adjustments in order to improve clarity), section 2.3 (extended the N-MDN summary), 3.1 (positive semi-definiteness), section 4 (extended introduction in order to give more space to our proposed approach and the overall structure of the section), section 5 (added a paragraph comparing our paper to Hug et al. in order to summarize novel aspects, extended comparison with Jorgensen & Osborne), Appendix (added positive semi-definiteness proof, added short model description to model comparison).

These adjustments are achieved mainly by either re-phrasing existing text or adding small bits of text in order to give either more information, context or structure. Most importantly, the main concepts, ideas and claims are not affected by these changes.

---

### Decision · Action_Editor_uYMP · 2023-11-27

**Recommendation:** Reject

**Comment:**

All reviewers recommended rejection at this stage of the manuscript. We would like to invite the authors to revise their paper significantly, taking the reviewer feedback seriously into account. Such a revision could then be resubmitted to TMLR at a later stage.

**Audience:**

The problem discussed in the paper is interesting and would certainly speak to at least a part of the TMLR audience.

**Claims And Evidence:**

All the reviewers agreed that the paper in its current form has major issues with its presentation, especially with the motivation of the method, its relationship to existing work, and the experimental results. As such, the reviewers were not convinced that the claims made by the paper are actually supported by the proper degree of evidence.

**Resubmission Of Major Revision:**

The authors may consider submitting a major revision at a later time.